# Application of Lévy Processes in Modelling (Geodetic) Time Series With Mixed Spectra

Jean-Philippe Montillet [1,2], Xiaoxing He [3], Kegen Yu [4], and Changliang Xiong [5]

[1] Physikalisch - Meteorologisches Observatorium Davos / World Radiation Center, Davos, Switzerland
[2] Space and Earth Geodetic Analysis laboratory (SEGAL), University Beira Interior, Covhila, Portugal
[3] School of Civil Engineering and Architecture, East China Jiaotong University, Nan Chang, China
[4] School of Environmental Science and Spatial Informatics, China University of Mining and Technology, Xuzhou, China
[5] Innovation Academy for Precision Measurement Science and Technology, Chinese Academy of Science (CAS), Xiaohongshan West Road, Wuhan, China

**Correspondence:** J.-P. Montillet (jpmontillet@segal.ubi.pt)

**Abstract.** Recently, various models have been developed, including the fractional Brownian motion (fBm), to analyse the stochastic properties of geodetic time series, together with the estimated geophysical signals. The noise spectrum of these time series is generally modelled as a mixed spectrum, with a sum of white and coloured noise. Here, we are interested in modelling the residual time series, after deterministically subtracting geophysical signals from the observations. This residual time series is then assumed to be a sum of three stochastic processes, including the family of Lévy processes. The introduction of a third stochastic term models the remaining residual signals and other correlated processes. Via simulations and real time series, we identify three classes of Lévy processes: Gaussian, fractional and stable. In the first case, residuals are predominantly constituted of short-memory processes. Fractional Lévy process can be an alternative model to the fBm in the presence of long-term correlations and self-similarity property. Stable process is here restrained to the special case of infinite variance, which can be only satisfied in the case of heavy-tailed distributions in the application to geodetic time series. Therefore, it implies potential anxiety in the functional model selection where missing geophysical information can generate such residual time series.

## 1 Introduction

Among the geodetic data, Global Navigation Satellite System (GNSS) time series have been of particular interest for the study of geophysical phenomenon at regional and global scales (e.g., study of the crustal deformation due to large Earthquakes, sea-level rise (Bock and Melgare , 2016; Herring et al. , 2016; He et al. , 2017)). This time series are the estimated daily position of the receiver coordinates. The position vector of a station can be decomposed in a geocentric cartesian axis system or in a local or topocentric cartesian axis system ($E,N,U$) in which the axes point east, north and up. These coordinates are influenced by the sum of three displacement modes (distinct classes of motion) that describes the progressive ground motion, any instantaneous jumps in position, and periodic or cyclical displacements. The progressive ground motion is generally refers as the tectonic rate. Jumps include coseismic offsets, which are real movements of the ground, and artificial offsets associated

with changes in the GNSS antenna and/or its radome, or changes in the antenna monument, etc. Nearly all GNSS time series exhibit a seasonal cycle of displacement which can be modelled as a Fourier series. These cycles are caused by seasonal changes in the water, snow and ice loads supported by the solid earth or (less commonly) by seasonal changes in atmospheric pressure. Therefore the model associated with each class of motion (or geophysical signals) is here defined as a functional model following Bevis and Brown (2014) and Montillet and Bos (2019) [Chapter 1]. Furthermore, these time series contain white noise and coloured noise. To model the different noise components, a stochastic noise model is defined. To name a few, it includes the First Order Gauss-Markov (FOGM) model, the white noise with power-law noise (Williams , 2003; Williams et al. , 2004), the Generalized Gauss Markov noise model (GGM), or the Band-pass noise (Langbein , 2008; Langbein and Svarc , 2019). (e.g., Flicker noise and white noise). The scientific community agrees with the existence of a trade-off in estimating both the stochastic and functional models (He et al. , 2017). More precisely, the choice of the stochastic model directly influences the estimation of the geophysical signals included in the functional model (i.e., tectonic rate, seasonal variations, slow-slip events (Bock and Melgare , 2016; He et al. , 2017)).

In addition, recent studies (Langbein and Svarc , 2019; He et al. , 2019) have also advocated the introduction of a random-walk to model small jumps and residual transient signals which is a non-stationary stochastic process. Thus, several studies (Montillet and Yu , 2015) proposed the use of the fractional Brownian motion (fBm), first developed by Mandelbrot et al. (1968), in order to model long-memory processes. Botai et al. (2011) and Montillet and Yu (2015) focused on modelling (residual) geodetic time series using the family of Lévy $\alpha$-stable distributions (Samorodnitsky and Taqqu , 1994; Nolan , 2018). The application of this family of distribution was supported by the ability to model long-memory processes and the existence of impulsive signals/noise bursts in the data sets suggesting deviations from Gaussian distribution (Botai et al. , 2011).

This work discusses several statistical assumptions (i.e. stationary properties, presence of long-term correlations) on the underlying processes in the GNSS time series, justifying the application of the fBm and the family of Lévy $\alpha$-stable distributions introduced by Montillet and Yu (2015). The Lévy stable distributions can model the heavy tail characteristics of some data sets with generally infinite variance. For example, the presence of unmodelled large jumps within the data can produce a distribution with large tails and infinite variance. In order to take into account a large variety of scenarios, we investigate and identify within the family of Lévy processes, which process can be applied to model geodetic time series.

Here, the statistical modelling is applied to residual time series following Montillet and Yu (2015). The residual time series are defined as the remaining time series after subtracting deterministically modelled tectonic rate and seasonal components (i.e. the functional model), from the GNSS observations. Therefore, our assumption is that the family of Lévy processes can model the remaining geophysical signals and correlations which have not been captured by the initial model used to produce the residual time series.

The next section starts with the statistical inference on the residual geodetic time series, including the application of the fBm model and the relationship with the Fractional Autoregressive Integrated Moving Average (FARIMA) model. Section 2.3 presents the assumptions on the use of the Lévy processes in the model of the residual time series. To do so, we model the residual geodetic time series as a sum of three stochastic processes, with the hypothesis that the third one is a Lévy process. It involves some justifications compared with established models in the scientific community. In Section 3, we develop an

$N$-step method based on the variations of the stochastic and functional models when varying the time series' length. This algorithm should verify our statistical assumptions on the third process. Section 3.1.1 and Section 3.1.2 focus on the application to simulated and real time series. Finally, Section 3.2 discusses the limits of modelling geodetic time series with Lévy processes.

## 2 The Stochastic Properties of the Residual Time Series and Statistical Inferences

### 2.1 Stochastic Modelling of Residual GNSS Time Series

Let us model the GNSS observations and residual time series as an additive model:

$$
\begin{aligned}
x_0(t_i) &= s_0(t_i) + n(t_i) \\
x(t_i) &= s_r(t_i) + n(t_i) \\
s_r(t_i) &= x_0(t_i) - s_0(t_i)
\end{aligned}
\tag{1}
$$

$x_0$ is the time series defined as the GNSS observations, $x$ the residual time series after subtracting the functional model ($s_0$). At each $i$-th epoch, $x(t_i)$ is a sum of a residual geophysical signal $s_r(t_i)$ and noise $n(t_i)$. Following Williams (2003) and He et al. (2017), the spectrum of the (residual) GNSS time series is best characterised by a stochastic process following a power-law with index $K$ (i.e. $P(f) = P_0(f/f_s)^K$, $f$ is the frequency, $P_0$ is a constant, $f_s$ the sampling frequency). A power-law noise model means that the frequency spectrum is not flat but is governed by long-range dependencies. An example is shown in Figure 3 using the $ASCO$ station, other examples are displayed in supplementary material. Power-law noise is a type of coloured noise. The coloured noise results from various parameters during the processing of the GNSS observations such as the mismodelling of GNSS satellites orbits, Earth orientation parameters, large-scale atmospheric or hydrospheric effects (Williams , 2003; Klos et al. , 2018). The stochastic noise model of the (residual) GNSS time series is then described with the variance:

$$
E\{\mathbf{n}^T\mathbf{n}\} = \sigma_{wn}^2 \mathbf{I} + \sigma_{pl}^2 \mathbf{J}(K)
\tag{2}
$$

where the vector $\mathbf{n} = [n(t_1), n(t_2), ..., n(t_L)]$ is a multivariate noise. At each epoch, we define $n(t_i) = n_{wn}(t_i) + n_{pl}(t_i)$, with $n_{wn}(t_i)$ and $n_{pl}(t_i)$ the white Gaussian noise (zero mean) and the power-law noise sample respectively. Note that this type of time series belongs to the family of mixed spectra, where the mixed spectrum results from the sum of the spectra corresponding to the two kinds of noise (Li , 2013). $T$ is the transpose operator, $\mathbf{I}$ the identity matrix, $\sigma_{pl}^2$ the variance of the power-law noise and $\mathbf{J}(K)$ the covariance matrix of the power-law noise ($K$ in $]0,2]$). The definition of $\mathbf{J}$ depends on the assumptions on the type of coloured noise (see supplementary material).

We estimate jointly the functional and stochastic models in order to produce $x$, based on a maximum likelihood estimator (MLE). To recall Montillet and Bos (2019) (Chapter 2), for linear models, the log-likelihood for a time series of length $L$ can be rewritten as:

$$
\ln(Lo) = -\frac{1}{2}\left[L\ln(2\pi) + \ln(\det(\boldsymbol{C})) + (\boldsymbol{x}_0 - \boldsymbol{A}\boldsymbol{z})^T \boldsymbol{C}^{-1}(\boldsymbol{x}_0 - \boldsymbol{A}\boldsymbol{z})\right]
\tag{3}
$$

This function must be maximised. Assuming that the covariance matrix $\boldsymbol{C}$ is known, then it is a constant and does not influence finding the maximum. $\boldsymbol{C}$ is here defined by Eq. (2). The term $(\boldsymbol{x}_0 - \boldsymbol{A}\boldsymbol{z})$ represent the observations minus the fitted model or $\boldsymbol{x}$ in Eq. (1). Note that $(\boldsymbol{A}\boldsymbol{z})$ is the matrix notation of $s_0$. The last term can be written as $\boldsymbol{x}^T\boldsymbol{C}^{-1}\boldsymbol{x}$ and it is a

quadratic function, weighted by the inverse of matrix $\boldsymbol{C}$. To select the geophysical model, and therefore estimate the associated parameters, one needs to consider carefully the location of the GNSS stations and the surrounding geodynamics. The model of $s_0$ is discussed in the supplementary material together with the software used to carry out the maximisation of $\ln(Lo)$. Note that the value of $L$ is here at least 9 years (3285 observations) in order to be able to model correctly the long-range dependency associated with the coloured noise and to detect slow transient signals according to He et al. (2019).

In the modelling of GNSS time series, a strong assumption is the so-called Gauss-Markov hypothesis (e.g., Montillet and Bos (2019) - *Chapter* 2) which states that the noise is Gaussian distributed. In order to keep applying the Gauss-Markov assumption on the noise observations of geodetic time series, we assume that the mean of the coloured noise is equal to $\mu_C(t)$, slowly varying with time. We then rule out the occurrence of specific events of large amplitude such as aggregations or burst of spikes (i.e. intermittency) which could invalidate such assumption (see Supplementary material for more information).

Moreover, if the probability density function of the noise is Gaussian or has a different density function with a finite value of variance, its fractal properties can be described by the Hurst parameter ($H$). The authors in Montillet et al. (2013) use the fBm in order to model the statistical properties of the residual time series. The essential features of this process are its self-similar behaviour, meaning that magnified and re-scaled versions of the process appear statistically identical to the original, together with its non-stationary property implying a never-ending growth of variance with time (Mandelbrot et al. , 1968). Previous

studies (e.g. (Mandelbrot et al. , 1968; Eke et al. , 2002)) showed that $H$ is directly connected with $K$ by the relation:

$$K = 2H - 1 \tag{4}$$

With this definition, flicker noise corresponds to $K$ equal to 1 or $H$ equal to 1, random walk to $K$ equal to 2 or $H$ equal to 1.5, and white noise to $K$ equal to 0 ($H$ equal to 0.5). Note that Eq. (4) is established for the fractional Gaussian noise according to (Eke et al. , 2002). The random-walk and the flicker noise are then classified as long-term dependency phenomena (Montillet

et al. , 2013).

Long-memory processes are modelled with a specific class of ARIMA models called FARIMA by allowing for non-integer differentiating. A comprehensive literature on the application of FARIMA can be found in financial analysis (Granger and Joyeux , 1980; Panas , 2001) and in geodesy (Li et al. , 2000; Montillet and Yu , 2015; Montillet and Bos , 2019). This model can generate long-memory processes based on the different values of the fractional index $d$ (Granger and Joyeux , 1980).

When $d$ equal to 0 it is an Autoregressive Moving Average (ARMA) process exhibiting short memory; when $-0.5 \leq d < 0$ the FARIMA process is said to exhibit intermediate memory or anti-persistence (Pipiras and Taqqu , 2017). This is very similar to the description of $H$ in the fBm. In the supplementary material, we recall the relationship between FARIMA, ARMA and fBm.

## 2.2 $\alpha$ Stable Random Variable and the Lévy $\alpha$-Stable Distribution

The fBm and the fractional Lévy distribution are well-known in statistics (Samorodnitsky and Taqqu , 1994) and in financial analysis (Panas , 2001; Wooldridge , 2010). The fractional Lévy distribution can model the Lévy processes and in particular the general family of $\alpha$ stable Lévy processes which can be self-similar and stationary (Samorodnitsky and Taqqu , 1994). Let us recall the definition of a stable random variable.

**Definition** *(Nolan , 2018), chap. 1, definition, 1.6* A random variable $X$ is stable if and only if $X \stackrel{d}{=} aZ + b$, where $0 < \alpha \le 2$, $-1 \le \beta \le 1$, $a \ne 0, b \in \mathbb{R}$ and $Z$ is a random variable with characteristic function $\phi(u) = E\{\exp(iuZ)\} = \int_{-\infty}^{\infty} \exp(iuz)F(z)$ $dz$. $F(z)$ is the distribution function of $Z$. $E\{.\}$ is the expectation operator. The characteristic function is:

$$\phi(u) = \begin{cases} \exp\left(-|u|^{\alpha}[1 - i\beta\tan\frac{\pi\alpha}{2}(\text{sign}(u))]\right), & \text{if } \alpha \ne 1 \\ \exp\left(-|u|[1 + i\beta\frac{2}{\pi}\text{sign}(u)]\right), & \text{if } \alpha = 1 \end{cases} \tag{5}$$

Where sign is the signum function, $\alpha$ is the characteristic exponent which measures the thickness of the tails of these distributions (the smaller the values of $\alpha$, the thicker the tails of distribution are), $\beta \in [-1, 1]$ is the symmetry parameter which set the skewness of the distribution. In general, no closed-form expression exists for these distributions, except for the Gaussian ($\alpha$ equal to 2), Pearson ($\alpha$ equal to 0.5, $\beta$ equal to $-1$) and Cauchy ($\alpha$ equal to 1, $\beta$ equal to 0) distributions.

Now, restricting to our case study, we assume that if the stochastic process exhibits a self-similar property, then it can be modeled by the fBm. Following (Weron et al. , 2005), the most commonly used extension of the fBm to the $\alpha$-stable case is the fractional Lévy stable motion (fLsm). The fLsm is $H$-self-similar and has stationary increments, with $H$ the Hurst parameter described before. Both the fBm and the fLsm follow an integral representation, with different properties of their kernel (see the supplementary material). The relationship between the fLsm reduces to the fBm when $\alpha = 2$. If $H = 1/\alpha$, we obtain the Lévy $\alpha$-stable motion which is an extension of the Brownian motion to the $\alpha$-stable case. Note that the Lévy $\alpha$-stable motion belongs to the Lévy processes.

## 2.3 The Residual Time Series Modelled as a Sum of Three Stochastic Processes

The residual time series is now modelled as a sum of three stochastic processes. Namely, it is the sum of a white noise, a coloured noise and a third process. It is a similar approach used in previous works (Langbein , 2008; Davis et al. , 2012; Langbein and Svarc , 2019; He et al. , 2019) looking at the presence of a random-walk component in the stochastic model, hence adding a third covariance matrix in Eq. (2). We postulate that this unknown stochastic process belongs to the Lévy processes, classified in three types depending on the assumptions on the underlying process:

1. (Lévy Gaussian) The Lévy process is a Gaussian Lévy process if the process follows the properties of a pure Brownian motion also called a Wiener process (identity variance matrix, zero-mean, stationary independent increment - (Haykin , 2002; Wooldridge , 2010)). That is the special case of the fLsm and fBm with $H = 1/2$. The residual time series is assumed to contain mostly short-term correlations modelled with an ARMA process. The residual time series should be modelled with a multivariate Gaussian distribution.

2. (Fractional Lévy) The residual time series exhibits self-similarity with possibly long-term correlations. The Fractional Lévy process is described by the model of the fLsm for the specific case reduced to the fBm. The long-term correlation process is mostly due to the presence of coloured noise (He et al. , 2017). As explained in Montillet and Yu (2015), the ratio of the amplitude of the coloured over white noise determines which stochastic model of the residual time series should be the most suitable between the FARIMA and ARMA processes. However, the Gauss-Markov assumption is still valid, therefore the residual time series should be modelled with a multivariate Gaussian distribution.

3. (Stable Lévy) The Lévy process is a Lévy $\alpha$-stable motion (not reduced to the fBm case). The Gauss-Markov assumption is not holding anymore. The distribution of the residual time series is potentially skewed, not symmetric, with possibly heavy tails, hence modelling with a Lévy $\alpha$-stable distribution. With the relationship between the Lévy $\alpha$-stable motion, the fBm and the FARIMA, we assume that the stochastic properties of the residual time series should be described with the FARIMA, especially in the presence of large amplitude coloured noise.

In the application to geodetic time series, the third case occurs mainly due to a misfit between the selected (stochastic and functional) model and the observations. Therefore, the residual time series withholds some remaining unmodelled geophysical signals or unfiltered large outliers which can potentially undermine the Gauss-Markov assumption (e.g., presence of heavy tails in the distribution of the residual time series). For example, if small jumps (or Markov jumps), outliers or other unknown processes are present, it results in a distribution of the residual time series not symmetric and with heavy tails. The functional model describing those jumps is a Heaviside step function (Herring et al. , 2016; He et al. , 2017) as shown in the appendices. In order to assume a Lévy $\alpha$-stable motion as the underlying stochastic model in geodetic time series, we restrict our assumption to small undetectable offsets, modelling them potentially as random-walk.

## 2.4 The N-Step Method

To recall Section 2.1, let us describe the functional model and the stochastic noise model described in Eq. (2) as a functional interpretation called $\mathcal{F}(\theta_1)$ and $\mathcal{G}(\theta_2)$. The functional model is the modeled geophysical signals, whereas the stochastic noise model described using the covariance matrix in Eq. (2) is equal to $\mathcal{G}(\theta_2)$. We define $\theta_1 = [a, b, (c_j, d_j)_{j=\{1,N\}}]$ and $\theta_2 = [a_{wh}, b_{pl}, K]$, the vector parameters for the functional and stochastic noise model respectively. For simplification, we have not included in the functional model the estimation of possible offsets in the time series (see appendices for the discussion). Also, $a_{wh}$ and $b_{pl}$ are the amplitude of the white and power-law noise respectively.

Furthermore, our method is based on varying the length of the time series resulting in the variations of the stochastic and functional models, which should allow classifying the type of Lévy process. The variations of the length of the time series should take into account that the coloured noise is a non-stationary signal (around the mean), and thus the properties (i.e. $b_{pl}$, $K$) vary non-linearly. However, varying the length of the time series over several years is not realistic taking into account that real time series can record undetectable transient signals, undocumented offsets and other non-deterministic signals unlikely to be modelled precisely (Montillet et al. , 2015). That is why we restrain the variations of the time series length to 1 year.

Let us call the geodetic time series $\mathbf{s}_1 = [s(t_1), ..., s(t_L)]$, $\mathbf{s}_2 = [s(t_1), ..., s(t_{L+1})]$ and $\mathbf{s}_N = [s(t_1), ..., s(t_{L+N})]$ at the first, second and $N$-th variation respectively. Note that the $N$ samples are equal to 1 year in this example, and for simplification we add only 1 sample at each step. That is not realistic, but the sole purpose is to be a pedagogical example. According to the functional notation above, the GNSS observations $\mathbf{s}$ and the estimated stochastic noise and functional models $\hat{\mathbf{s}}$ are equal to:

$$
\begin{aligned}
\mathbf{s} &= \mathcal{F}(\theta_1) + \mathcal{G}(\theta_2) \\
\hat{\mathbf{s}} &= \mathcal{F}(\hat{\theta_1}) + \mathcal{G}(\hat{\theta_2})
\end{aligned}
\tag{6}
$$

Let us describe the method for the first, second and $N$-th step such as:

$\underline{1^{st} \ step:}$

$$
\begin{aligned}
\mathbf{s}_1 &= [s(t_1), ..., s(t_L)] \ (\text{Time} \quad \text{Series}) \\
[\hat{\theta_1}]_1, [\hat{\theta_2}]_1 &= \underset{\theta_1, \theta_2}{argmax}\{\mathbf{s}_1 - (\mathcal{F}(\theta_1) + \mathcal{G}(\theta_2))\} \\
\hat{\mathbf{s}}_1 &= \mathcal{F}([\hat{\theta_1}]_1) + \mathcal{G}([\hat{\theta_2}]_1) \ (\text{Estimated} \quad \text{model}) \\
\Delta\mathbf{s}_1 &= \mathbf{s}_1 - \mathcal{F}([\hat{\theta_1}]_1) \ (\text{residual} \quad \text{T.S.}) \\
&\simeq \mathcal{G}([\hat{\theta_2}]_1) + \epsilon_1
\end{aligned}
$$

$\underline{2^{nd} \ step:}$

$$
\begin{aligned}
\mathbf{s}_2 &= [s(t_1), ..., s(t_{L+1})] \ (\text{Time} \quad \text{Series}) \\
[\hat{\theta_1}]_2, [\hat{\theta_2}]_2 &= \underset{\theta_1, \theta_2}{argmax}\{\mathbf{s}_2 - (\mathcal{F}(\theta_1) + \mathcal{G}(\theta_2))\} \\
\hat{\mathbf{s}}_2 &= \mathcal{F}([\hat{\theta_1}]_2) + \mathcal{G}([\hat{\theta_2}]_2) \ (\text{Estimated} \quad \text{model}) \\
\Delta\mathbf{s}_2 &= \mathbf{s}_2 - \mathcal{F}([\hat{\theta_1}]_2) \ (\text{residual} \quad \text{T.S.}) \\
&\simeq \mathcal{G}([\hat{\theta_2}]_2) + \epsilon_2
\end{aligned}
$$

$\underline{N-th \ step:}$

$$
\begin{aligned}
\mathbf{s}_N &= [s(t_1), ..., s(t_{L+N})] \ (\text{Time} \quad \text{Series}) \\
[\hat{\theta_1}]_N, [\hat{\theta_2}]_N &= \underset{\theta_1, \theta_2}{argmax}\{\mathbf{s}_N - (\mathcal{F}(\theta_1) + \mathcal{G}(\theta_2))\} \\
\hat{\mathbf{s}}_N &= \mathcal{F}([\hat{\theta_1}]_N) + \mathcal{G}([\hat{\theta_2}]_N) \ (\text{Estimated} \quad \text{model}) \\
\Delta\mathbf{s}_N &= \mathbf{s}_N - \mathcal{F}([\hat{\theta_1}]_N) \ (\text{residual} \quad \text{T.S.}) \\
&\simeq \mathcal{G}([\hat{\theta_2}]_N) + \epsilon_N
\end{aligned}
\tag{7}
$$

where $\hat{}$ corresponds to the estimated vector or observations. $[.]_j$ means the $j$-th iteration of the estimated quantity. $\Delta\mathbf{s}_j$ is the residual time series at the $j$-th step. $\epsilon_j$ (with $j$ in $[1, 2, .., N]$) is the unmodelled signals and stochastic processes at the $j$-th step.

Note that the methodology requires the estimation of the parameters of the functional and stochastic noise models $[\hat{\theta_1}]_j, [\hat{\theta_2}]_j$

via MLE as described in Section 2.1 in the maximization of $\ln(Lo)$ in Eq. (3) (see also Hector software Bos et al. (2013) in the supplementary material).

To recall the assumptions in Section 2.3, the residual time series $\Delta \mathbf{s}_N$ is modelled as a sum of three stochastic processes corresponding to the white noise, coloured noise and a Lévy process. Using $N$ iterations/steps and our statistical inferences on the Lévy processes (i.e., Lévy Gaussian, Fractional Lévy and Stable Lévy), we make several assumptions on the estimated

parameters and selected stochastic models in order to characterize the third process. Table 1 summarises these assumptions. We use specific mathematical symbols to differentiate between them. $\triangleq$ means the equality in terms of distribution. $\simeq$, $\sim$ and $\neq$ are related to the variations of the estimated parameters of the stochastic model associated with the first and the $N$-th step. The symbol $\simeq$ means that there are little differences (less than $3\%$) between the estimated parameters of the stochastic noise model associated with the first and the $N$-th iteration. The symbol $\sim$ means that we allow bigger differences up to $20\%$. With

much larger values, we use the symbol $\neq$. Note that the variation of the estimated stochastic noise parameters $[\hat{\theta_2}]_j$ between the first and the $j$-th step is calculated using the sum of the difference in absolute value between the estimates (e.g., between the first and $j+1$ step, $||[\hat{\theta_2}]_1 - [\hat{\theta_2}]_{j+1}||$). We deduce a percentage of the variations based on the sum in absolute value of the estimates $[\hat{\theta_2}]_1$.

**Table 1.** Assumptions on the functional model and the stochastic parameters estimated via $N$ iterations (see,$N$-Step method) to characterize the type of Lévy processes within the geodetic time series. The symbols and notations are explained in Section 2.4

| Type of Process | Lévy Gaussian | Fractional Lévy | Stable Lévy |
|---|---|---|---|
| Mathematical Assumptions | $\mathcal{G}([\hat{\theta_2}]_1) \simeq \mathcal{G}([\hat{\theta_2}]_N)$ $\mathcal{F}([\hat{\theta_1}]_1) \simeq \mathcal{F}([\hat{\theta_1}]_N)$ | $\mathcal{G}([\hat{\theta_2}]_1) \sim \mathcal{G}([\hat{\theta_2}]_N)$ $\mathcal{F}([\hat{\theta_1}]_1) \sim \mathcal{F}([\hat{\theta_1}]_N)$ | $\mathcal{G}([\hat{\theta_2}]_1) \neq \mathcal{G}([\hat{\theta_2}]_N)$ $\mathcal{F}([\hat{\theta_1}]_1) \neq \mathcal{F}([\hat{\theta_1}]_N)$ |
| (Distribution) $\Delta^1 \mathbf{s} \triangleq$ | Gaussian | Gaussian | Lévy $\alpha$-stable |
| Model To Characterize Processes | ARMA(p,q) | ARMA(p,q) or FARIMA(p,d,q) | FARIMA(p,d,q) |

Furthermore, the fitting of the ARMA($p$,$q$) and FARIMA($p$,$d$,$q$) model to the residual time series is carried out by maximum-

likelihood following Sowell (1991). The lags $p$ and $q$ vary within the interval $[0,5]$. Also, the selection of the model which best fits the residual time series, is performed by minimizing the Bayesian Information Criterion (BIC) following Montillet and Yu (2015). Finally, one can wonder if the anxiety in the model selection ARMA, FARIMA) in presence of heavy-tails can modify the performance of the BIC. This topic is currently debated in the statistical community (e.g., (Panahi , 2016)). Large tails should be detected in the fitting of the Lévy $\alpha$-stable distribution. Various methods exist to estimate the parameters of this

distribution (Koutrouvelis , 1980), however we use the maximum-likelihood method of Nikias and Shao (1995). Due to the direct relationship between the index $\alpha$ and $H$ recalled in Section 2.1, we assume that the amplitude of the coloured noise is higher than the white noise, therefore the FARIMA should be chosen de facto over the ARMA model.

## 3   Lévy Processes Applied to Geodetic Time Series Analysis

This section deals with the application of the N-step algorithm to simulated and real time series. This approach should verify our statistical inferences formulated in Section 2.3. Note that the simulations of the GNSS time series are comprehensively explained in the supplementary material (supplement $D$).

### 3.1   Application to Simulated and Real Time Series

We have restrained our simulations to the stochastic model corresponding to the flicker noise (with white noise - $FN+WN$) and power-law (with white noise - $PL+WN$). In addition to simplify our study, we have preliminary applied the method based on the Akaike information criterion developed in He et al. (2019) on the real time series to select the optimal stochastic noise model. Therefore we have selected real time series with stochastic models $FN+WN$ and $PL+WN$. We are not going to develop further this topic in this study, but readers can refer to He et al. (2019).

#### 3.1.1   Simulated Time Series

We simulate $10$ years long time series fixing $a_{wh}$ to 1.6 mm; the tectonic parameters $a$ varying between $[1-3]$ mm/yr and $b$ equal 0; and the seasonal signal with only the first harmonic $(c_1, e_1)$ equal to $(0.4, 0.2)$ mm/yr. Details on the noise simulations are given in the supplementary material. According to Table 1, we vary the amplitude of coloured noise $b_{cl}$ following three scenarios:

  A. low value (i.e. $b_{cl} < 0.1$ mm/$yr^{K/4}$)

  B. intermediate value (i.e. $1mm/yr^{K/4} > b_{cl} > 0.1$ mm/$yr^{K/4}$)

  C. high value (i.e. $1mm/yr^{K/4} < b_{cl} < 4mm/yr^{K/4}$)

Note that in the scenario $C$, the process is unlikely zero-mean stationary. Also, it is mentioned when $K$ is equal to 1 (flicker noise) or $1.5$ (power-law noise) in the simulations of the coloured noise.

Figure 1a, 1b and 1c display the results when averaging over $50$ time series. The variations are in steps of $[0, 0.3, 0.5, 0.7, 0.8, 1]$ year (see X-axis). We show both the variations of the stochastic and functional models. The Y-axis displays the variations of the models in terms of percentage as discussed in the previous section.

The first result which is common to all three figures, is that the variations in terms of variance of the functional model increases faster than for the results associated with the stochastic model. Previous studies have shown that there is a part of the noise amplitude absorbed in the functional model (Williams , 2003; Montillet et al. , 2015). In our scenario, the estimation of the linear trend may fit partially into the power-law noise, hence reducing the variations of the stochastic noise model. This effect can be amplified with higher spectral indexes (Montillet and Bos , 2019). Now, Figure 1 shows that over 1 year the variations of the stochastic and functional models are less than $4\%$ (on average) for small amplitude coloured noise, whereas when increasing the coloured noise amplitude the variations increase quickly (e.g., more than $20\%$ for the large coloured noise

**Figure 1.** Percentage of variations of the estimated parameters included in the stochastic and functional models when varying the length of the time series. The letters $(A)$, $(B)$ and $(C)$ refer to the various scenarios with different coloured noise amplitudes.

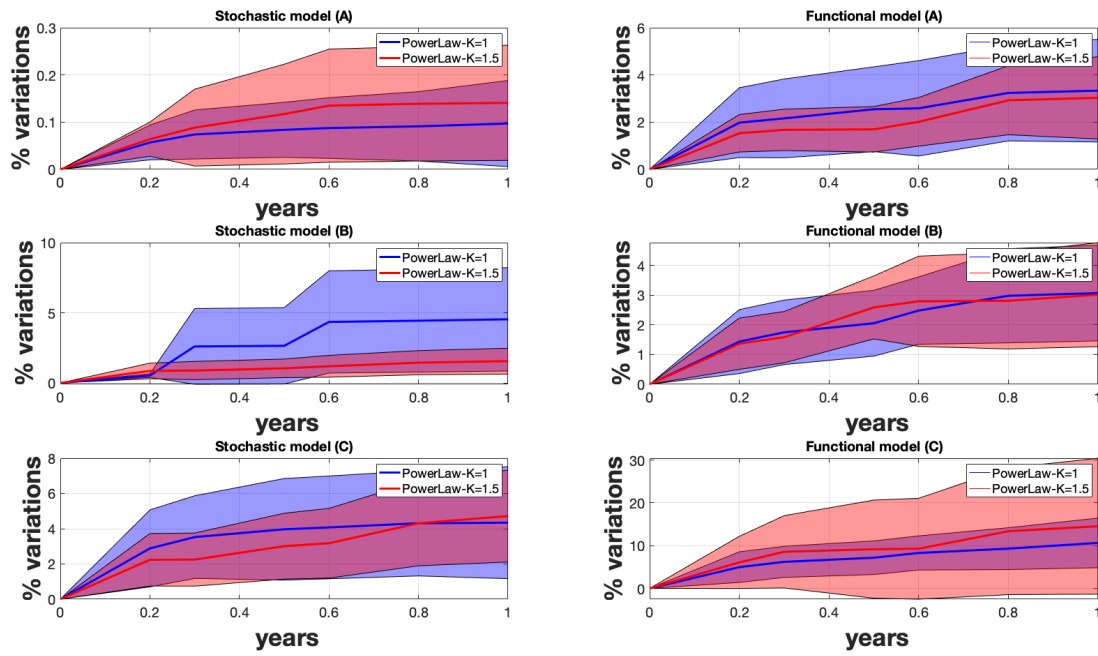

amplitude corresponding to the scenario $(C)$). We assume that part of the large variations of the coloured noise is wrongly absorbed in the estimation of the functional model.

**Table 2.** Statistics on the error when fitting the ARMA and FARIMA model to the residual time series following the three scenarios

| Error (mm) | | case $A$ | case $B$ | case $C$ |
|---|---|---|---|---|
| | $K$ | $b_{cl} < 0.1$ mm/$yr^{K/4}$ | $1mm/yr^{K/4} > b_{cl} > 0.1$ mm/$yr^{K/4}$ | $1mm/yr^{K/4} < b_{cl} < 3mm/yr^{K/4}$ |
| ARMA | 1.0 | $1.44 \pm 0.01$ | $1.74 \pm 0.01$ | $1.89 \pm 0.04$ |
| | 1.5 | $1.46 \pm 0.01$ | $1.76 \pm 0.04$ | $1.95 \pm 0.05$ |
| FARIMA | 1.0 | $1.91 \pm 0.02$ | $1.85 \pm 0.02$ | $1.46 \pm 0.02$ |
| | 1.5 | $1.89 \pm 0.01$ | $1.75 \pm 0.03$ | $1.59 \pm 0.05$ |

Now, Table 2 shows the standard deviation of the difference ($Mean\ Square\ Error$) between the ARMA /FARIMA model and the residuals (i.e. $\mathbf{res}_i$ in Eq. (7)). We do not display any mean, because the fit of the models are done on the zero-mean residuals. Note that the value is averaged over the $50$ simulations, together with the variations of the length of the time series described above. The table also displays the averaged correlation between the distribution of the residuals and the Normal or Lévy $\alpha$-stable distribution. In agreement with the theory, we can see that the ARMA model fits well residuals

**Table 3.** Correlation between the distribution of the residuals and the Normal (*Corr. Normal* ) distribution or the Lévy $\alpha$-stable distribution (*Corr. Lévy* ) and the Anderson-Darling test (AD) following scenario $A$, $B$ and $C$. The results of the AD test is the probability over the 50 trials of accepted null hypothesis. [*Lévy* ] or [*Normal* ] means the type of distribution uses as the null hypothesis

| *Corr.* $[0-1]$ | $K$ | case $A$ $b_{cl} < 0.1$ mm/$yr^{K/4}$ | case $B$ $1mm/yr^{K/4} > b_{cl} > 0.1$ mm/$yr^{K/4}$ | case $C$ $1mm/yr^{K/4} < b_{cl} < 3mm/yr^{K/4}$ |
|---|---|---|---|---|
| *Corr. Normal* | 1.0 | $0.93 \pm 0.04$ | $0.92 \pm 0.06$ | $0.92 \pm 0.04$ |
|  | 1.5 | $0.92 \pm 0.04$ | $0.91 \pm 0.04$ | $0.91 \pm 0.05$ |
| *Corr. Lévy* | 1.0 | $0.92 \pm 0.05$ | $0.94 \pm 0.04$ | $0.96 \pm 0.03$ |
|  | 1.5 | $0.93 \pm 0.03$ | $0.94 \pm 0.03$ | $0.95 \pm 0.03$ |
| *AD test* [Normal] | 1.0 | $0.98 \pm 0.01$ | $0.96 \pm 0.01$ | $0.94 \pm 0.03$ |
|  | 1.5 | $0.97 \pm 0.01$ | $0.96 \pm 0.02$ | $0.93 \pm 0.04$ |
| *AD test* [Lévy] | 1.0 | $0.97 \pm 0.02$ | $0.97 \pm 0.01$ | $0.97 \pm 0.03$ |
|  | 1.5 | $0.98 \pm 0.01$ | $0.97 \pm 0.02$ | $0.97 \pm 0.02$ |

with small amplitude coloured noise ($b_{cl}$), whereas with the increase of $b_{cl}$ the FARIMA model fits better than the ARMA model. Looking at Table 3 in terms of correlation, the Lévy $\alpha$-stable distribution fits as good as the Normal distribution as long as the distribution of the residuals is Gaussian without large tails or asymmetry. That is why the Anderson-Darling test accepts the two distributions when the residual time series is Gaussian distributed without tails. In Section 2, we emphasized that the family of Lévy $\alpha$-stable distributions includes the Normal distribution with specific values for the parameters of the characteristic function (see Eq. (5)). Thus, the results show that for the amplitude of coloured noise corresponding to scenario $B$ (i.e. Intermediate - in Table 2 and 3), the two distributions show similar results. However, the scenario $C$ can potentially generate some aggregation processes in the simulated time series and introducing an asymmetry or large tails in the distribution of the residuals, therefore it emphasizes that the family of Lévy $\alpha$-stable distributions perform the best in modelling the residuals' distribution. To further support this result, we have added the Anderson-Darling test (Anderson and Darling , 1952) in order to test for the large tails in the distribution of the residuals in Table 3. Following our previous development, we have used the Normal and the Lévy $\alpha$-stable distributions as null hypothesis. The results show that this test detects mostly large tails for the scenario $C$ which corresponds to when the family of Lévy $\alpha$-stable distributions perform better than the Normal distribution.

Finally, those three scenarios support the theory where in the case of small amplitude coloured noise, the stochastic noise properties are dominated by the Gaussian noise, hence defining a third process as a Gaussian Lévy . However, the increase of the coloured noise amplitude shows that it is much more difficult to discriminate between the fractional Lévy and the stable Lévy . The results indicate that the third process can be modelled as a stable Lévy process when mostly the FARIMA fits the residuals due to large amplitude long-memory processes, hence creating a heavy-tail distribution. This result is restrictive for the application to geodetic time series.

**Figure 2.** Percentage of variations of the estimated parameters included in the stochastic and functional models when varying the length of the daily position GNSS time series corresponding to the stations $DRAO$, $ASCO$ and $ALBH$. The statistics are estimated over the East and North Coordinates

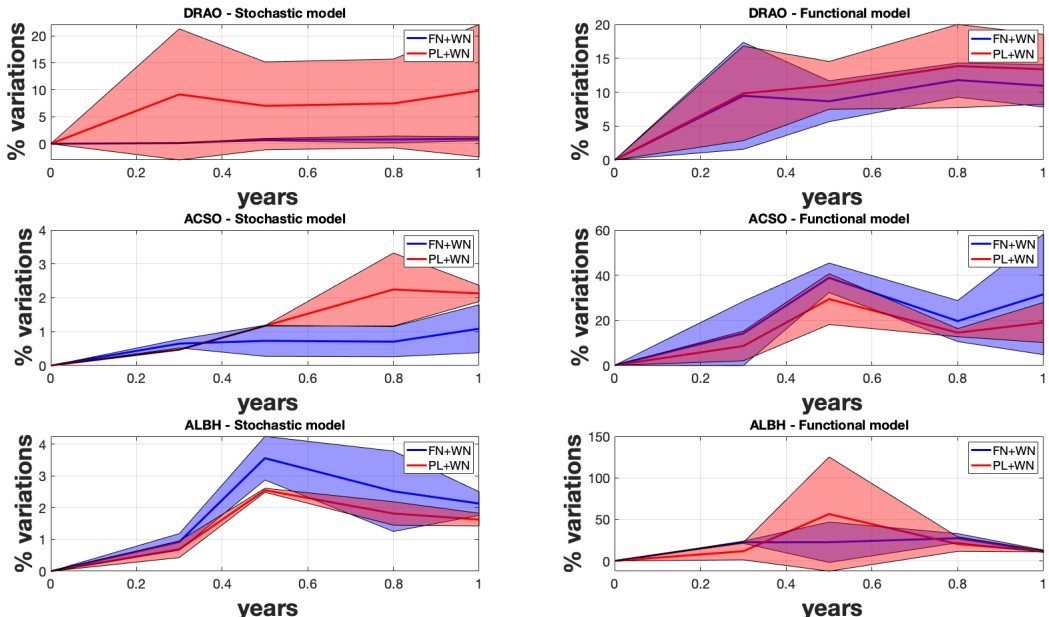

 ### 3.1.2 Real Time Series

We process the daily position time series of three GNSS stations namely $DRAO$, $ASCO$ and $ALBH$ retrieved from the UNAVCO website (UNAVCO , 2009). The functional model includes the tectonic rate, the first and second harmonic of the seasonal signal, and the occurrence time of the offsets. This occurrence time is obtained from the log file of each station. However, $ALBH$ is known to record slow-slip events from the Cascadia subduction zone (Melbourne et al. , 2005). Thus, we include the offsets provided by the Pacific Northwest Geodetic Array (Miller et al. , 1998). In this scenario we do not know which stochastic model could fit the best the observations. Thus, we use two models: the $PL + WN$ together with the $FN + WN$. Note that in appendices, we display the time series of some of the coordinates, together with the processing and the fitting of the distributions.

Similar to the previous section, Figure 2 displays the percentage of variations of the stochastic and functional models averaged over the East and North coordinates of each station. Note that the average over the three coordinates is displayed in the appendices (see Figure A1). Because the Up coordinate contains much more noise than the East and North coordinates (Williams et al. , 2004; Montillet et al. , 2013), it amplifies the variation of both stochastic and functional models.

Looking at Figure 2, the first result is that for all the stations, there is a strong dependence with the selected noise model. When selecting the power-law noise over the flicker noise model, there is an additional variable to estimate (i.e. the power-law

noise exponent, $K$, in Eq. (4) ) within the stochastic noise model. This dependence is already discussed in previous studies (He et al. , 2017, 2019).

The second result is the large variations of the functional model compared with the stochastic model. To recall the simulation results, the functional model partially absorbs the variations of the noise, i.e. the tectonic rate partially fits into the power-law noise. In addition, to some extend at $ASCO$, the sudden increase in the functional model variations at $0.5$ year may be explained due to the absorption of some of the noise with the second harmonic of the seasonal signal.

When comparing the variations of the stochastic and functional models with amplitude below $20\%$ for $DRAO$ and $ASCO$, the results agree with the definition of the fractional Lévy process defined in Table 1 as third process modelling the residuals of the East and North components. The variations of the functional model associated with $ALBH$ are much larger than the other two stations, especially for the $PL+WN$ model with variations up to $50\%$. Those large variations can be explained due to the slow-slip events and the difficulty to model the post-seismic relaxations between two consecutive events He et al. (2019).

**Table 4.** Statistics on the Error when fitting the ARMA and FARIMA model to the residual time series for each coordinate of the stations $ALBH$, $DRAO$ and $ASCO$ based on the $PL+WN$ stochastic noise model. Correlation between the distribution of the residuals and the Normal (*Corr. Normal* ) and the Lévy $\alpha$-stable distributions (*Corr. Lévy* ). The last column is the Anderson-Darling test. [*Lévy* ] or [*Normal* ] means the type of distribution uses as the null hypothesis (1 accepted, 0 rejected)

| DRAO | (err. in mm) ARMA | (err. in mm) FARIMA | Corr. Normal | Corr. Lévy | AD test [Lévy ] | AD test [Normal ] |
|---|---|---|---|---|---|---|
| East | $1.07 \pm 0.01$ | $1.10 \pm 0.07$ | 0.94 | 0.97 | 1 | 1 |
| North | $1.02 \pm 0.02$ | $1.01 \pm 0.01$ | 0.96 | 0.96 | 1 | 1 |
| Up | $2.32 \pm 0.21$ | $2.15 \pm 0.30$ | 0.97 | 0.98 | 1 | 1 |
| ASCO | | | | | | |
| East | $0.77 \pm 0.01$ | $0.77 \pm 0.06$ | 0.98 | 0.97 | 1 | 1 |
| North | $0.84 \pm 0.03$ | $0.73 \pm 0.03$ | 0.97 | 0.96 | 1 | 1 |
| Up | $2.71 \pm 0.12$ | $2.34 \pm 0.17$ | 0.92 | 0.96 | 1 | 0 |
| ALBH | | | | | | |
| East | $0.97 \pm 0.06$ | $0.87 \pm 0.06$ | 0.98 | 0.98 | 1 | 1 |
| North | $1.54 \pm 0.03$ | $1.06 \pm 0.14$ | 0.97 | 0.98 | 1 | 1 |
| Up | $4.36 \pm 0.17$ | $4.08 \pm 0.25$ | 0.92 | 0.95 | 1 | 0 |

Furthermore, Table 4 displays the statistics on the error when fitting the ARMA and FARIMA models to the residuals estimated with the $PL+WN$ stochastic noise model. Figure 3 shows the time series $ASCO$ for the East coordinate using the full time series. Note that Table A1 displays in the appendices the results when using the $FN+WN$ stochastic noise model. The FARIMA and ARMA models perform closely for all three stations. The large value for the Up coordinate is due to the amplitude of the noise much larger for this coordinate than for the East and North components. In terms of correlating the distribution of the residuals with the Normal and the Lévy $\alpha$-stable distribution, the correlation value is relatively the same for all stations which indicates that the distribution of the residuals are Gaussian with the absence of large tails. The Anderson-

Darling test also confirms this result when the acceptance of the null hypothesis is the same for the two distributions. Those results further support the selection of the fractional Lévy process as the third stochastic process. However, the study of real time

series also underlines the difficulty to characterize statistically this third stochastic process. Note that the Anderson-Darling test shows also that there are some variations for the up coordinate where the Lévy $\alpha$-stable distribution is only selected. As discussed above, the noise on the up coordinate is much larger than in the other coordinates, therefore it may create small tails.

## 3.2  Discussion on the Limits of Modelling with Lévy Processes

In Montillet and Yu  (2015), it was assumed that the infinite variance of the residual time series comes from large tails of

330 the distribution (i.e heavy tails), generated by a large amplitude of coloured noise, outliers and other remaining geophysical signals. The same study implied that the values of the noise variance should be bounded, excluding extreme values. This is an important assumption to decide whether or not (symmetric) Lévy $\alpha$-stable distributions can be used to model any geodetic time series. This section investigates how the variance due to residual tectonic rate or seasonal signal evolves with the length of the residual time series (i.e. $L$ epochs).

To recall Section 2.1 and the assumption on the noise properties, let us develop the close-form formula of the mean and variance of the residual time series. The residual time series is $\Delta \mathbf{s}_1 = [\Delta s_1(t_1), ..., \Delta s_1(t_L)]$ as defined in Eq. (7). The mean $\langle \Delta s_1(L) \rangle$ and variance $\sigma^2(L)$ are computed over $L$ epochs (i.e. considering the $L$-th epoch defined as $t_L = Ldt$, with the sampling time $dt$ equal 1 for simplification and without taking into account any data gaps in order to have a continuous time series). Based on Papoulis and Unnikrishna Pillai  (2002), one can estimate $\langle \Delta s_1(L) \rangle$ in the cases of a remaining linear trend

such as:

$$
\begin{aligned}
\Delta s_1(t_i) &= a_r t_i + b_r + n(t_i) \\
\langle \Delta s_1(L) \rangle &= \frac{1}{L} \sum_{i=1}^{L} (a_r t_i + b_r + n(t_i)) \\
&= b_r + a_r \frac{(L+1)}{2} + \mu_C \\
&\simeq a_r \frac{L}{2} + \mu_C
\end{aligned}
\tag{8}
$$

where $a_r$ and $b_r$ are the amplitude and the intersect of the remaining tectonic rate. Note that the subscript $r$ designates *residual* of a geophysical signal in the remaining. $\simeq$ is the approximation for $L \gg 1$. The variance $\sigma^2(L)$ is equal to:

$$
\begin{aligned}
\sigma^2(L) &= \frac{1}{L} \sum_{i=1}^{L} (\Delta s_1(t_i) - \langle \Delta s_1(L) \rangle)^2 \\
&= a_r^2 \frac{(L+1)(2L+1)}{6} - a_r^2 \frac{(L+1)^2}{4} + b_r^2 + \frac{2a_r}{L} Cross(a_r, n) + \sigma_n^2(L) - \mu_C(\mu_C + a_r(L+1)) \\
&\simeq \frac{a_r^2 L^2}{12} + \sigma_n^2(L) + b_r^2 - \mu_C a_r L
\end{aligned}
\tag{9}
$$

Note that $Cross(a_r, n)$ is the cross term between $a_r t_i$ and the noise term $n(t_i)$. Now, if we assume that the remaining seasonal signal $S_r(t)$ is a pseudo periodic function at frequencies similar to the seasonal signal, hence taking the form

**Figure 3.** GNSS time series for the $ASCO$ station (East coordinate) with the $PL+WN$ model. A/ the time series together with the functional model, B/ the power-spectrum, C/ Residual time series with Lévy $\alpha$-stable distribution, D/ cumulative density function residual time series and Lévy $\alpha$-stable distribution (Corr. Lévy $= 0.98$), E/ Residual time series with Normal distribution, F/ cumulative density function residual time series and Normal distribution (corr. Norm. $= 0.97$ ).

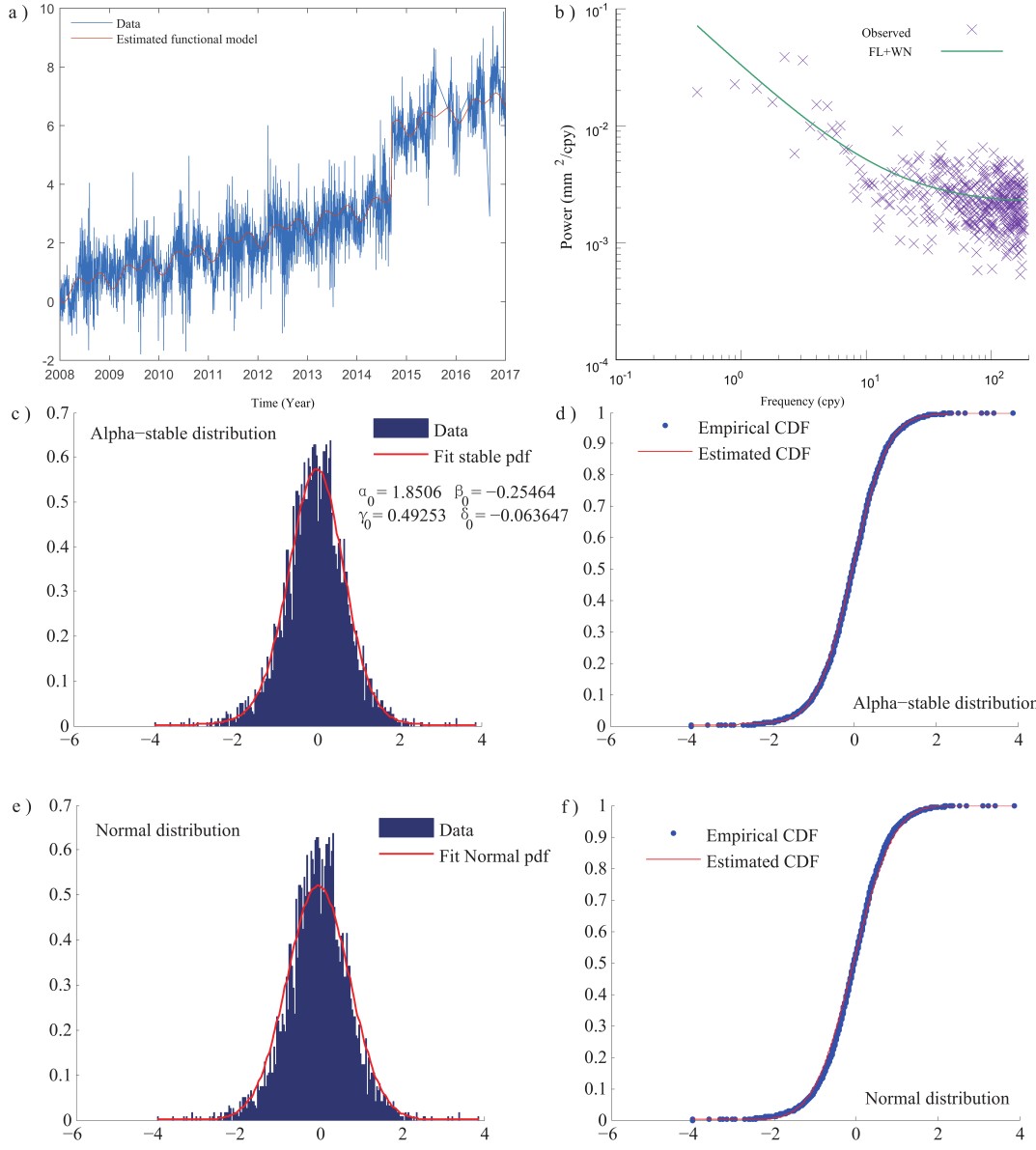

$S_r(t) = \sum_{j=1}^{N} c_{r,j} \cos(d_j t) + e_{r,j} \sin(d_j t)$. Thus, we can do the same estimation as above in the case of a remaining pseudo periodic component in the residual time series, such as:

$$
\begin{aligned}
\Delta s_1(t_i) &= S_r(t_i) + n(t_i) \\
\langle \Delta s_1(L) \rangle &= \frac{1}{L} \sum_{i=1}^{L} (S_r(t_i) + n(t_i)) \\
&\simeq \delta + \mu_C
\end{aligned}
\tag{10}
$$

where $\delta$ is the average of the remaining seasonal signal. It is assumed to be independent of $L$ and bounded such as a periodic function. The variance is equal to:

$$
\begin{aligned}
\sigma^2(L) &= \frac{1}{L} \sum_{i=1}^{L} \sum_{j=1}^{N} c_{r,j}^2 \cos(d_j t)^2 + e_{r,j}^2 \sin(d_j t)^2 + \sigma_n^2(L) \\
&\quad + \frac{2}{L} Cross(S_r, n) - \langle \Delta s_1(L) \rangle^2 \\
&\simeq \sigma_n^2(L) + \sum_{j=1}^{N} c_{r,j}^2 + e_{r,j}^2 - (\delta + \mu_C)^2
\end{aligned}
\tag{11}
$$

with $Cross(S_r, n)$ is the cross term between $S_r(t)$ and $n(t)$. For all the cross terms, we assume that the deterministic signals and the noise are completely uncorrelated, which is valid with white Gaussian noise (e.g., signal processing - Papoulis and Unnikrishna Pillai (2002)). As previously discussed in Section 2.1, coloured noise is characterised by long-memory processes, hence producing non-zero covariance with residual signals. Due to the varying amplitude of the coloured noise in geodetic time series with mixed spectra, the uncorrelated assumption is currently debated within the community (Herring et al. , 2016; He et al. , 2017). Therefore, recent works have introduced a random component together with a deterministic signal: nonlinear rate (Wang et al. , 2016; Dmitrieva et al. , 2017), non-deterministic seasonal signal (Davis et al. , 2012; Chen et al. , 2015; Klos et al. , 2018). Thus, strictly speaking, the estimate $\sigma^2$ should be seen as an upper bound.

The closed-form solution of the variance $\sigma^2(L)$ shows that the variance is unbounded in the case of a residual linear trend. If this residual trend originates from various sources not well-described in the functional and stochastic model (i.e. undetected jumps, small amplitude random-walk component) of the geodetic time series, the amplitude of this trend should be rather small ($a < 1$ mm/yr) considering the length of GNSS time series available until now ($L < 30$ years). Unless this nonlinear residual trend has a large amplitude, a correction of the functional model must be done a posteriori due to possible anxiety between the models and the observations. The same remarks can be applied to the variance of the remaining seasonal signal where a large amplitude would imply a misfit with the functional model. Thus, we expect rather small amplitude of the coefficients $c_{r,j}$ and $e_{r,j}$ (e.g., $c_{r,j} \sim 0.1$ mm to $e_{r,j} \sim 0.001$ mm). Also, in the appendices, we have developed a similar formula to take into account undetected offsets, where we show that the variance is also bounded. In this case, a large value would mean that one or several large offsets have not been included in the functional model.

## 4  Conclusions

We have investigated the statistical assumptions behind using the fBm and the family of Lévy $\alpha$-stable distributions in order to model the stochastic processes within the residual GNSS time series. We model the residual time series as a sum of three stochastic processes. The first two processes are defined from the stochastic model and assumptions on the noise properties of the geodetic time series. The third process is assumed to belong to the Lévy processes. We then distinguish three cases. In the case of a residual time series containing only short-term processes, the process is a Gaussian Lévy process. In the presence of long-term correlations and exhibiting self-similarity property, fractional Lévy processes can be seen as an alternative model of using the fBm. Due to the linear relationship between the Hurst parameter and the fractional parameter of the FARIMA, it is likely that the FARIMA can fit the residual time series under specific conditions (i.e. amplitude of the coloured noise). The third case is the stable Lévy process, with the presence of long-term correlations, high amplitude aggregation processes or random-walk.

In order to check our model, we have simulated mixed spectra time series with various levels of coloured noise. We have then developed a $N$-step methodology based on varying the length of the time series to study the variations of the stochastic and functional models and also to model the distribution of the residuals. The results emphasize the difficulty to separate the fractional Lévy process and the stable Lévy process mainly due to the absorption of the variations of stochastic processes by the functional model, unless the distribution of the residuals exhibits heavy-tails.

The discussion on the limits of modelling the stochastic properties of the residuals with the stable Lévy process underlines that the infinite variance property can only be satisfied in the case of heavy-tailed distributions, resulting from 1/ the presence of a large amplitude random-walk (e.g., temporal aggregation in financial time series), 2/ an important misfit between the models (i.e. functional and stochastic) and the observations, which means that there is anxiety in the choice of the functional model (e.g., unmodelled large jumps, large outliers). With longer and longer time series, one may be able to statistically characterize more precisely the third stochastic process. Finally, future work should investigate the autoregressive conditional heteroscedasticity (ARCH) model applied to GNSS time series in order to model differently the stochastic properties (e.g. non-stationarity beyond the mean).

*Acknowledgements.*  We would like to thank Dr. Machiel S. Bos from the SEGAL - University of Beira Interior for multiple discussions on the stochastic properties of the GNSS time series. He also develops the Hector software used in this study. Dr. Xiaoxing He would like to acknowledge the Natural Science Foundation of Jiangxi Province supporting his work on GNSS time series (Research on the Noise Model Refinement Algorithm on GNSS Time Series with Non-linear Variation). Finally, we would like to thank the reviewers for the constructive comments helping to improve this manuscript.

## Appendix A: Estimation of the Variance in the Presence of Offsets

Here, we model the offsets in the time series as Heaviside step functions according to He et al. (2017). Following Section 3.2, the residual time series in presence of remaining offsets can be written such as:

$$\Delta s_1(t_i) = \sum_{k=1}^{ng} g_k \mathcal{H}(t_i - T_k) + n(t_i) \tag{A.1}$$

Where $\mathcal{H}$ is the Heaviside step function; $g_k$ is the amplitude of the offset; $T_k$ is the time of occurrence of the offset; $ng$ is the number of offsets; $n$ is the noise. One can estimate the average over $L$ epochs:

$$\begin{aligned}
\langle \Delta s_1(L) \rangle &= \frac{1}{L} \sum_{i=1}^{L} \left( \sum_{k=1}^{ng} g_k \mathcal{H}(t_i - T_k) \right) + \mu_C(t) \\
&= \frac{1}{L} \sum_{k=1}^{ng} g_k \mathcal{H}(t_L - T_k) + \mu_C(t)
\end{aligned} \tag{A.2}$$

Note that $\mu_C(t)$ is the mean of the coloured noise, slowly varying in time (see Section 2.1). The variance is equal to:

$$\begin{aligned}
\sigma^2(L) &= \frac{1}{L} \sum_{i=1}^{L} \left( \sum_{k=1}^{ng} g_k \mathcal{H}(t_i - T_k) + n(t_i) - \langle \Delta s_1(L) \rangle \right)^2 \\
&\simeq \sigma_n^2(L) + \frac{1}{L} \left( \sum_{k=1}^{ng} g_k \mathcal{H}(t_L - T_k) \right)^2 - (\langle \Delta s_1(L) \rangle)^2
\end{aligned} \tag{A.3}$$

In the presence of small (undetectable) offsets ( $g_k < 1$ mm), we can further assume that $\langle \Delta s_1(L) \rangle \sim \mu_C(t)$ and $\sigma^2(L) \sim \sigma_n^2(L) - \mu_C^2(t)$. For multiple large uncorrected offsets (i.e. noticeable above the noise floor), the variance can be large, but the distribution of the residual time series should look like multiple Gaussian distributions overlapping each other corresponding to the segments of the time series defined by those noticeable offsets. This case is not taken into account in our assumptions summarized in Table 1, because it supposes that there is a large anxiety about the chosen functional model (i.e. obviously miss-

ing some large noticeable offsets well above the noise floor). Note that for a comprehensive discussion about offset detection, we invite readers to refer to Gazeaux et al. (2013) and He et al. (2017).

## Appendix B: Additional Tables and Figures

**Table A1.** Statistics on the Error when fitting the ARMA and FARIMA model to the residual time series for each coordinate of the stations $ALBH$, $DRAO$ and $ASCO$ based on the $FN + WN$ stochastic noise model. Correlation between the distribution of the residuals and the Normal (*Corr. Normal* ) and the Lévy $\alpha$-stable distributions (*Corr. Lévy* ). The last column is the Anderson-Darling test. [*Lévy* ] or [*Normal* ] means the type of distribution uses as the null hypothesis (1 accepted, 0 rejected)

| DRAO | (err. in mm) ARMA | (err. in mm) FARIMA | Corr. Normal | Corr. Lévy | AD test [Lévy ] | AD test [Normal ] |
|---|---|---|---|---|---|---|
| East | $1.07 \pm 0.01$ | $1.00 \pm 0.02$ | 0.95 | 0.95 | 1 | 1 |
| North | $1.02 \pm 0.02$ | $1.32 \pm 0.07$ | 0.96 | 0.98 | 1 | 1 |
| Up | $2.33 \pm 0.18$ | $2.20 \pm 0.32$ | 0.94 | 0.96 | 1 | 1 |
| ASCO | | | | | | |
| East | $0.77 \pm 0.01$ | $0.75 \pm 0.07$ | 0.95 | 0.96 | 1 | 1 |
| North | $0.85 \pm 0.03$ | $0.74 \pm 0.05$ | 0.94 | 0.96 | 1 | 1 |
| Up | $2.18 \pm 0.14$ | $2.51 \pm 0.21$ | 0.93 | 0.94 | 1 | 1 |
| ALBH | | | | | | |
| East | $0.97 \pm 0.04$ | $0.86 \pm 0.06$ | 0.95 | 0.95 | 1 | 1 |
| North | $1.52 \pm 0.08$ | $1.08 \pm 0.10$ | 0.96 | 0.95 | 1 | 1 |
| Up | $3.83 \pm 0.21$ | $3.32 \pm 0.15$ | 0.93 | 0.95 | 1 | 0 |

**Figure A1.** Percentage of variations of the estimated parameters included in the stochastic and functional models when varying the length of the daily position GNSS time series corresponding to the stations $DRAO$, $ASCO$ and $ALBH$. The statistics are estimated over the East, North and Up Coordinates

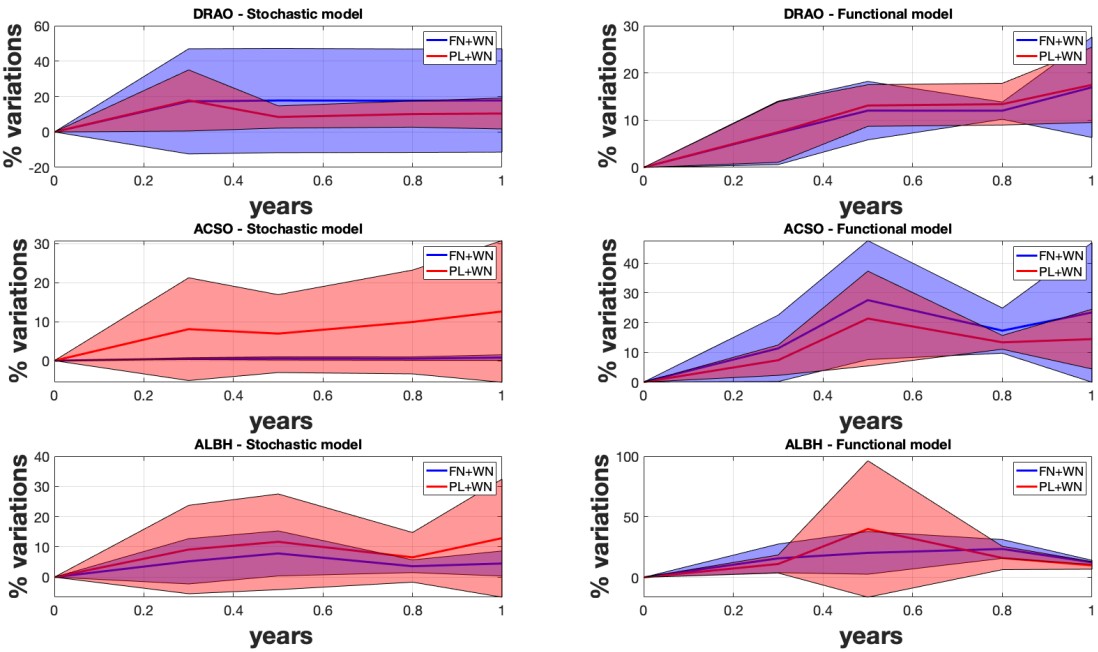

**Figure A2.** GNSS time series for the $DRAO$ station (North coordinate) with the $FN+WN$ model. A/ the time series together with the functional model, B/ the power-spectrum, C/ Residual time series with Lévy $\alpha$-stable distribution, D/ cumulative density function residual time series and Lévy $\alpha$-stable distribution (Corr. Lévy $= 0.98$), E/ Residual time series with Normal distribution, F/ cumulative density function residual time series and Normal distribution (Corr. Norm. $= 0.96$).

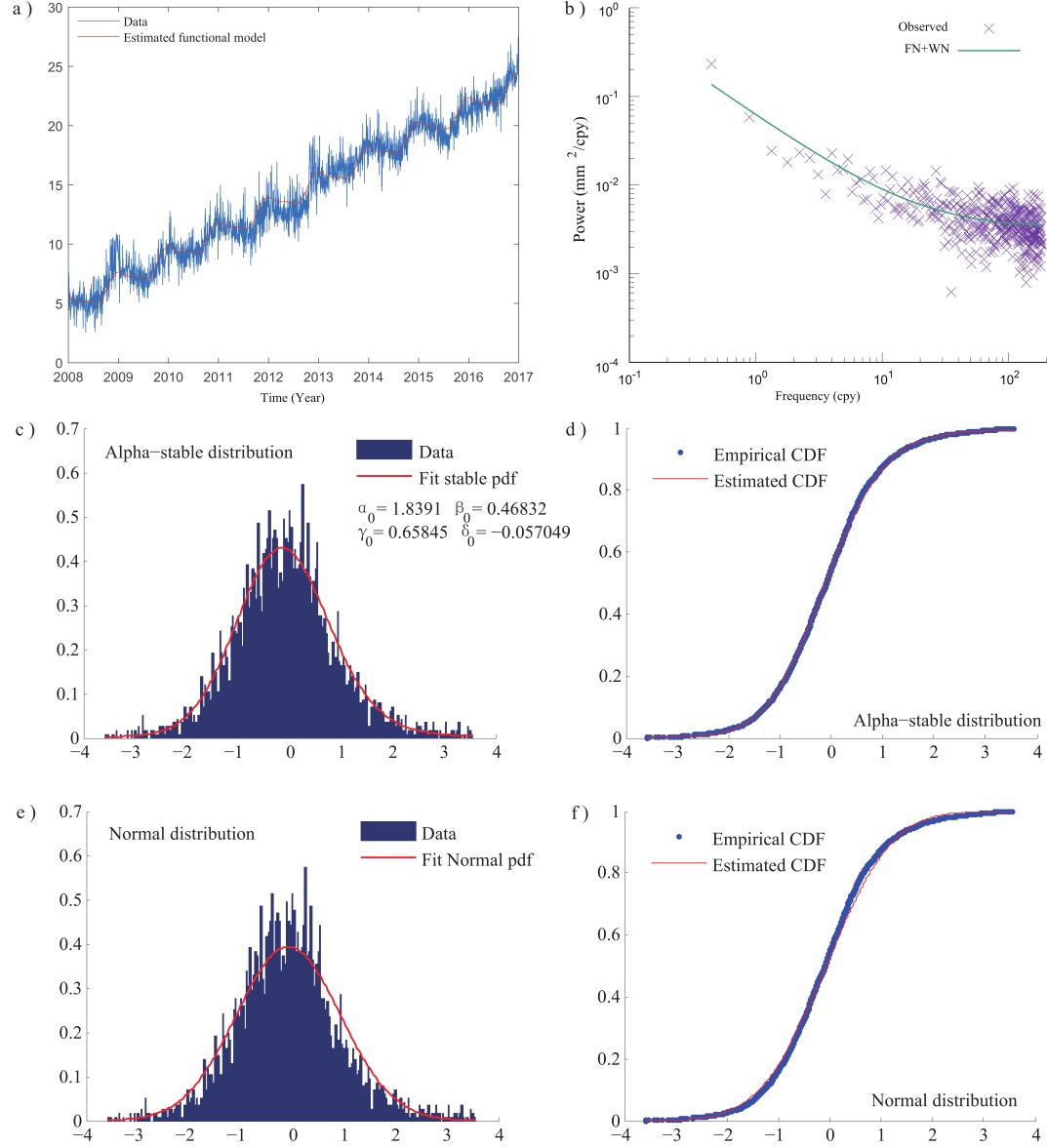

**Figure A3.** GNSS time series for the $ALBH$ station (East coordinate) with the $PL+WN$ model. A/ the time series together with the functional model, B/ the power-spectrum, C/ Residual time series with Lévy $\alpha$-stable distribution, D/ cumulative density function residual time series and Lévy $\alpha$-stable distribution (Corr. Lévy $= 0.98$), E/ Residual time series with Normal distribution, F/ cumulative density function residual time series and Normal distribution (Corr. Norm. $= 0.98$).

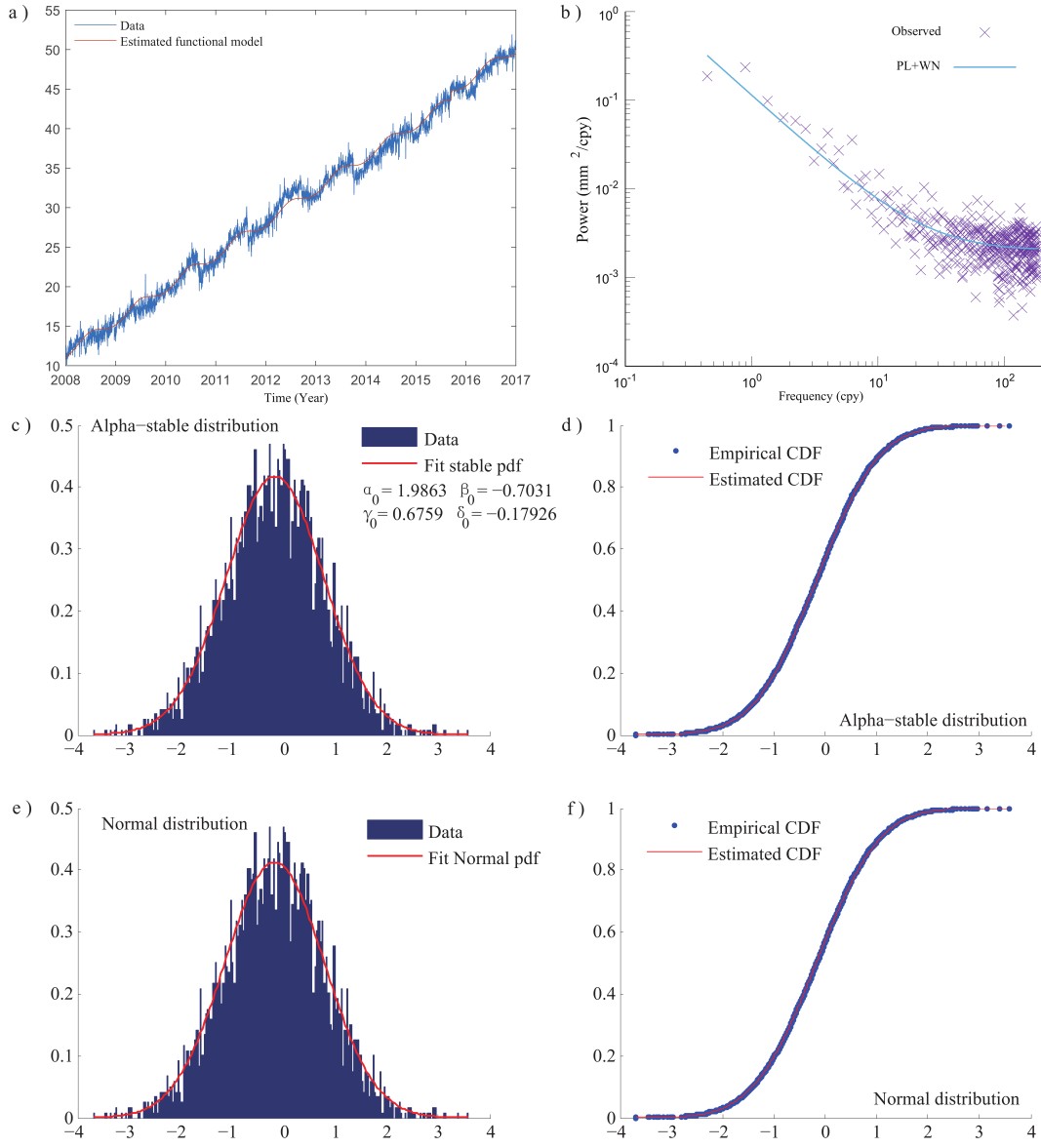

**Figure A4.** GNSS time series for $DRAO$ (Up coordinate) with the $PL+WN$ model. A/ the time series together with the functional model, B/ the power-spectrum, C/ Residual time series with Lévy $\alpha$-stable distribution, D/ cumulative density function residual time series and Lévy $\alpha$-stable distribution(Corr. Lévy $= 0.98$), E/ Residual time series with Normal distribution, F/ cumulative density function residual time series and Normal distribution(Corr. Norm. $= 0.97$).

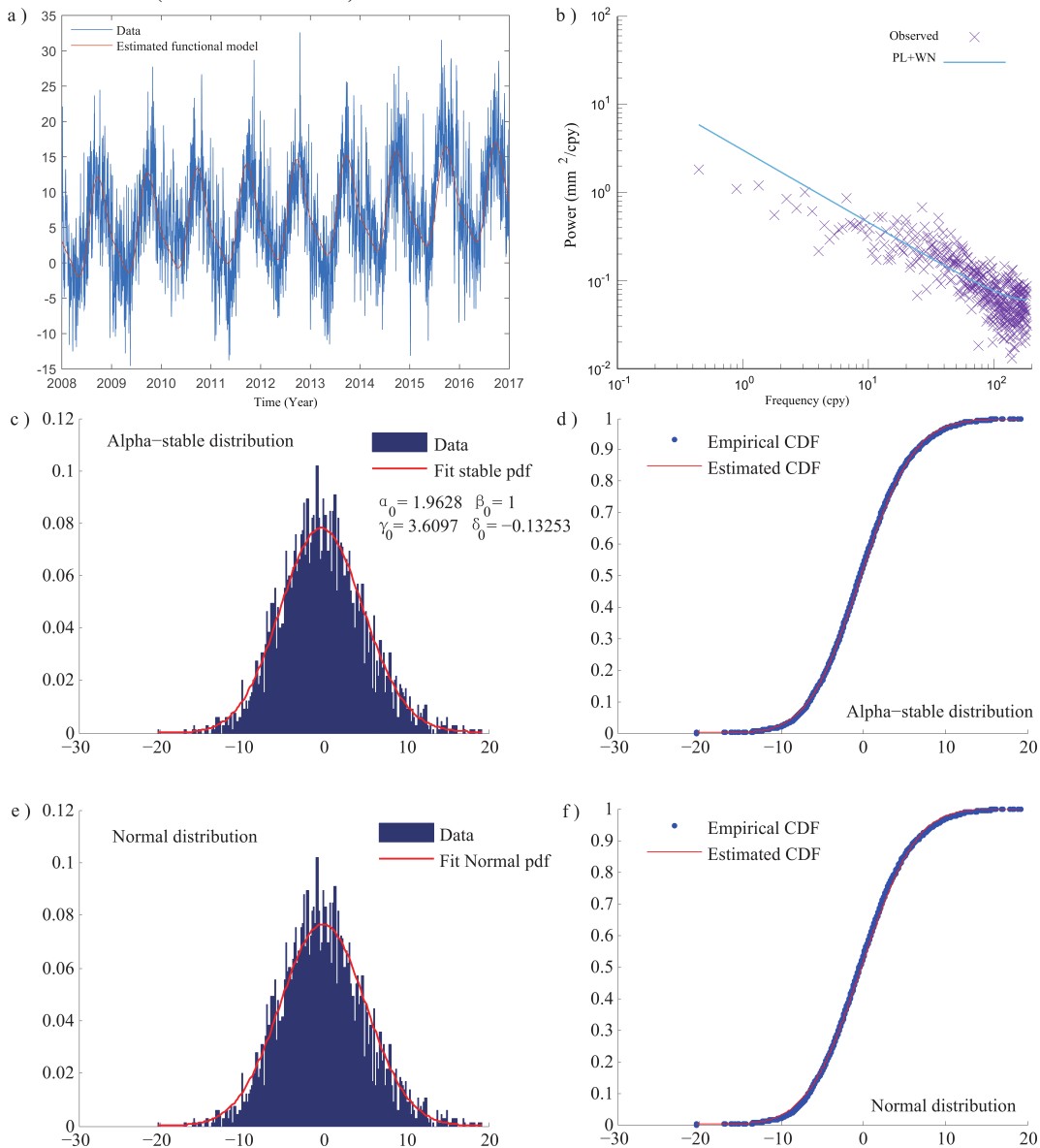

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
