# Peer review of "Application of Lévy Processes in Modelling (Geodetic) Time Series With Mixed Spectra"

_Nonlinear Processes in Geophysics, 2020_

## Referee Comment (RC1) · Anonymous Referee #1 · 23 Jul 2020

Current manuscript is a revised manuscript, and it has been significantly been made better. However, many issues are still left, especially with the writing style, and this makes the text unreadable.

However, I think that this time a major revision is sufficient.

I have two main criticisms, and these affect the results section, which should be rewritten after the basic work is done.

1. The mathematical model is really messily written, and it is hard to understand it. Take all the definitions from the appendix, and put them in the text, and remove all the speculative/descriptive material. It is totally impossible to understand as the model is

not explicitly defined. Something like:

" Let us model a GNSS observations as an additive model

$$x(t) = s_r(t) + n(t) \tag{1}$$

" ... and then define the different choices for $s_r(t)$ and $n(t)$, and also write explicitly what their corresponding discrete versions are.

2. After you have done the basic model, you can construct an a posteriori distribution of the unknown parameters, and then the target is to sample these parameters with MCMC methods or obtain optimisation-based MAP/ML estimators. Define explicitly your estimators with respect to posterior distribution. Of course you can use some other constructions as well, but define your estimators explicitly. The $N$-step method is totally heuristic and should not be included in the manuscript. Please come up with some mathematics/stats-based parameter estimation algorithm.

Some minor comments:

p. 2, line 27, long-memory processes (i.e. coloured noise). – Coloured noise is not necessarily long-memory process.

p. 2, line 43 – Gaussian distribution is part of the Lévy distribution, be precise.

p. 3 Equation (1) – Explain the whole model here, it is impossible to guess what the model is, write the formula out explicitly. What is $s_0$? What is $s_r(t)$? Is noise $n(t)$ a continuous-time variable? What is the mean of $n(t)$?

p. 3 Equation(3), please don't use $\mathbf{J}$ as a full covariance matrix – it is often used to denote identity matrix. Use something very distinctive, like $\mathbf{C}$.

p. 3: "Note that the length of the geodetic time series (L) considered in this study is at least 9 years ( 3285 observations)." Can you plot this data?

p. 3: "which states that the noise is Gaussian distributed, therefore n follows a multivariate Gaussian probability density function. " ... this is a tautology.

p. 3: Line 82 – You have not defined $\beta$, and for all $\beta < 0$, the matrix $\mathbf{J}(\beta)$ does not exist. Be more precise.
* * *

---

## Short Comment (SC1) · 3 Aug 2020

Dear Reviewer,

Thank you for your comments. We will emphasize some key points that perhaps have not been sufficiently underlined in the manuscript (and will be further emphasized in the next revised version).

1- "The mathematical model is really messily written, and it is hard to understand it.Take all the definitions from the appendix, and put them in the text, and remove all the speculative/descriptive material."

[R] Here, we have rewritten the model and the explanation in order to avoid the same issue as in the previous version. We want to emphasize that we are working on residual time series which are defined in Appendix A:

"to obtain the residual time series x(t) ( t the epoch), we subtract the functional model "s0(t)" to the GNSS observations "s(t)".The functional model of the geophysical signals is based on the polynomial trigonometric method (Li et al. , 2000; Williams ,2003; Tregoning and Watson , 2009) . . . "

In addition, we have added a comprehensive discussion on how to obtain the residual time series. It includes the ML estimator (using Hector) which is used to jointly estimate the stochastic noise model and the geophysical parameters. The geophysical parameters are described with the equation A.1 (Appendix). Now, in the residual time series described in Eq. 1, we state that "s_r" is the residual geophysical signal. Therefore, it includes 1/ the residual signal from the estimation of the model in equation A.1 2/ the missing signals which have been omitted in the geophysical models such as small short-time duration transient signals, small amplitude exponential relaxation, small offsets (undetectable by eyes) . . . We must emphasize that this mismodelled geophysical signals (especially the small offsets) can be modelled/assimilated as random-walk in the stochastic noise model. That is one of the justifications behind using three stochastic processes (instead of 2) in modelling the stochastic noise component of the GNSS time series (see L. 135 ".. It is similar to the approach used in previous works (Langbein , 2008; Davis et al. , 2012;Langbein and Svarc , 2019; He et al. , 2019) looking at the presence of a random-walk component in the stochastic model, hence adding a third covariance matrix in Eq. (3). . . ."). This section will be revised accordingly with most of the Appendix A moved to section 2.1. Note that to formulate the model of the residual time series, we use the model of the GNSS observations defined in (Montillet and Bos, 2019: Chapter 1 and 2). Also the covariance matrix defined in Eq. (3) is also based on (Montillet and Bos, 2019: Chapter 2). That will be clarified in the next version of the article.

All the material is here to give the background of this work, and justification of the assumptions (especially the consideration of the third stochastic process). It is worth mentioning that this material has been substantively revised and superfluous discussion/descriptions have been deleted from the previous submitted article.

2- " After you have done the basic model, you can construct an a posteriori distribution of the unknown parameters, and then the target is to sample these parameters with MCMC methods or obtain optimisation-based MAP/ML estimators. Define explicitly your estimators with respect to posterior distribution. Of course you can use some other constructions as well, but define your estimators explicitly. The N-step method is totally heuristic and should not be included in the manuscript. Please come up with some mathematics/stats-based parameter estimation algorithm."

[R] The goal of this work is to understand/investigate the use of three stochastic processes instead of 2. To do that, we use the Levy processes which can characterise this third stochastic process. Table 1 displays the three cases (Gaussian, Fractional and Stable Levy) together with the assumptions. Basically, there is nothing heuristic in the choice of these 3 cases and the statistical assumptions. We have comprehensively documented past works which lead to these assumptions in Section 2.3.

The justification behind the N-step method is that we want to investigate the variations in the stochastic noise models in order to classify the third stochastic process. Thus, we make the assumptions that the properties of the noise can vary with the length of the time series, because the coloured noise is not stationary. We cannot vary the time series with a length of more than 1 year, because this value is a trade-off between too short and too long, meaning that with only a few months the noise property remains stationary, and at the opposite with longer time we risk including more geophysical signals due to events affecting the stations (Earthquake, land subsidence ...). The N-step method is carried out using Hector (based on ML) in order to estimate jointly the stochastic noise parameters and the geophysical signals. However, the characterisation of the third stochastic process is not only based on the results with the N-step

method, but also by fitting the residual time series with the Levy alpha stable distribution, and testing which model ARMA vs FARIMA fits the residuals. Note that the fitting of the Levy alpha stable distribution is done by using a ML approach (see line 220). The fitting of the ARMA and FARIMA models follow the method (based on ML) developed in Montillet and Yu (2015). Thus, the characterisation of this third stochastic process is not heuristic and it is based on testing the residual time series where the N-step method is the backbone of our methodology.

Thank you for the minor comments, it will help to improve the manuscript further. All the necessary corrections will be included in the next draft.

---

## Referee Comment (RC2) · Reik Donner (Referee) · 20 Sep 2020

The authors discuss the stochastic model based description of time series with mixed spectra motivated by the typical properties of geodetic (GNSS) time series. They consider this problem in terms of an additive model with two or three components representing different types of stochastic processes, among one is taken from the family of Lévy processes in three possible flavors. The authors describe a procedure for the step-wise iterative estimation of the associated process parameters and apply their approach to both, artificial data and three real-world GNSS series.

While I find that the overall topic is relevant and appropriate for the readership of Non-

linear Processes in Geophysics, my impression is that the specific background of the models considered here needs to be more clearly presented for a broader audience. Generally, the overall model structure should be motivated in a more systematic and more transparent way than done in the present version of the manuscript. Some specific questions I came across when working through this discussion paper, which I suggest the authors to briefly address in the process of revising especially Sections 1 and 2.1 of the manuscript, include the following:

Can you explain a bit more systematically how (and why) different types of non-stationarity are associated to the different components of your stochastic model? While I think that this is relatively clear for "deterministic" (monotonic) trends and seasonality, abrupt offsets due to jumps in the series could either be associated with the functional (deterministic offsets due to seismic events or instrumentation changes) or the stochastic part (in the latter case, I would expect that they can be considered as Lévy flights, but I am not sure if such effects commonly occur in GNSS series). Moreover, I am wondering about non-stationarity beyond just the mean, i.e., possible time-dependence in the variance (or even higher-order distributional characteristics), which I believe cannot be captured by the present model setup but would require either some multiplicative component (e.g., functional model times noise of a certain type) or a stochastic model component beyond ARMA/FARIMA that inherets the property of conditional heteroscedasticity (e.g., some ARCH/GARCH type model). Again: I do neither claim that those characteristics are necessarily typical for GNSS series (but the manuscript title points to a wider class of time series with mixed spectra where this could arise), nor do request the authors to provide a solution for any possible type of situation that could arise. What I however would appreciate to see more transparently is the list of assumptions that underlie the discussed class of stochastic models, and some brief discussion on if and how possible phenomenological findings (like heteroscedasticity of the noise component, intermittency,...) are included in the general model structure studied in the present work.

You mention the use of the Hector software at several, in my opinion not necessarily relevant places. Since you do not provide any specific information that is unique to this software, I recommend mentioning this only in the acknowledgements and removing corresponding unnecessary statements elsewhere (e.g. ll. 200-201, 390, 402-403, 417-418).

Regarding the analysis of the modelling results, I am not convinced that it makes sense statistically to use correlations as a measure for the goodness-of-fit/matching between an empirical and a theoretical distribution. For the purpose of this study (Tab. 3/4 and associated text), some two-sample statistical test like Kolmogorov-Smirnov or Anderson-Darling (probably rather the latter since heavy-tailed distributions are included) appear more reasonable, while it – one the one hand - is unclear how the correlations have been computed and – on the other hand – the mean correlation values are trivially very high while the provided uncertainty margins (Tab. 3/4) clearly exceed the possible range of correlation values (bounded from above by one) and are therefore meaningless.

Regarding the general presentation, the manuscript is well readable but contains a quite relevant number of minor grammatical errors, mainly confusion of singular and plural forms, missing or wrong prepositions, and other minor things that I do not wish to list here explicitly. Thorough proofreading during the revision process is recommended.

Some more specific suggestions on helpful additions and minor modifications to the text are listed below:

Ll.19/20: "ground motion" sounds to me more appropriate than "soil motion", unless you explicitly aim to focus on just the uppermost soil layer of the ground

L.75: I don't quite see that it is relevant to mention the length of the real-world series at this point – to me this rather fits in the results section.

Ll.78-80: Can you explain the relationship between slowly varying mean of colored

noise and the Gauss-Markov assumption a bit more explicitly?

Ll.89-90 or below: Maybe you could add a brief comment on the relevance of H for processes with infinite variance (e.g. Cauchy-class processes)?

Ll.101-107: In the context of this brief discussion of FARIMA models, you might also briefly recall the relationship between the fractional model order d and Hurst exponent H.

L.114: The definition includes a variable k that I don't find appearing anywhere before.

Ll.154-155: This sentence would benefit for some further explanation for non-specialists.

L.171: In which sense do you consider colored noise to be non-stationary?

Figures 1 and 2: emphasize on the different ranges on the y axes somewhere in the figure caption

L.243: The list of values given in the text is inconsistent with that shown in the figure.

L.246: I would not speak of "earlier" here due to the low number of data points in the figure, but rather refer to the overall values of the variance.

L.262: What do you mean by "driving parameters"?

Ll.271-274: I understand this as that heavy tails in the series can either be attributed to the residuals or to the third component. Can such an attribution be actually unique?

L.325 and several times later: There are quite a few cases of equations spanning over different lines with duplicate left-hand sides indicating that those are in fact different equations even though they are not.

L.348: "colored noise can generate long-memory processes" seems a bit odd to me; rather, colored noise commonly constitutes a long-memory process

L.349: Doesn't the mentioned varying amplitude of the colored noise rather call for

multiplicative/heteroscedastic models? (See main comment above.)

L.373: "long-term correlations" (remove "processes")

You should make use of the full functionality of the LATEX template; e.g. use \appendix followed by \section to generate individual appendices. Note that the references must not appear as an appendix. Moreover, also the use of \citet versus \citep in the text could be improved.

Ll.417-430: Please emphasize that the expressions in Z denote composite operators in terms of the backshift operator Z applied to x(t) and b(t), respectively.

L.434: instead of the term "hyperbolic", "algebraic" seems more commonly used

L.436-439: The link to fBm is a bit unclear here. Since you consider stationary processes (FARIMA class), it would be more reasonable in my opinion to link this to fractional Gaussian noise (fGn) the aggregation of which than provides sample paths for fBm.

L.451: extend the equation by an expression including f_t(u) as the latter is used in the text below

Appendix F: Did you consider putting those materials into an Electronic Supplementary Material instead of an Appendix in the main paper?

―――――――――――

---

## Author Comment (AC1) · 28 Sep 2020

Dear Reviewer,

Thank you for your comments. We would like to give a few answers to some of your comments. The modifications that we intend to do directly to the manuscript are highlighted with (R).

Major comments

Âń Can you explain a bit more systematically how (and why) different types of non-stationarity are associated to the different components of your stochastic model? Âż

The stochastic noise model is basically divided in 2 components white + coloured noise. The coloured noise results from various parameters during the processing of the GNSS observations such as the mismodelling of GNSS satellites orbits, Earth orientation parameters, large-scale atmospheric or hydrospheric effects . . . (see Williams 2003, Williams 2004, Klos et al., 2018). Several studies in the past 2 decades have advocated that the coloured noise component is best described as a power-law noise, and particularly as a Flicker noise (power-law exponent = 1). A full discussion can be found in Montillet and Bos, 2019 (chapter 2).

(R): To recall why the coloured noise is modelled in the stochastic noise model, we will add the sentence "The coloured noise results from various parameters during the processing of the GNSS observations such as the mismodelling of GNSS satellites orbits, Earth orientation parameters, large-scale atmospheric or hydrospheric effects . . . (see Williams 2003, Williams 2004, Klos et al., 2018)."

- About the length of the time series

As we mentioned in the manuscript, there are several stochastic noise models which have been used to model the properties of the power-law noise (and in particular the Flicker noise), it includes FOGM, GGM . . . see line 25-30. The coloured noise is slowly varying, therefore the length of the time series is an important parameter in order to model this type of noise.

(R): The text will be modified accordingly (see also the discussion in the minor comments).

- "I am wondering about non-stationarity beyond just the mean, i.e., possible time-dependence inthe variance (or even higher-order distributional characteristics), which I believe can-not be captured by the present model setup but would require either some multiplicative component (e.g., functional model times noise of a certain type) or a stochasticmodel component beyond ARMA/FARIMA that inherets the property of conditional heteroscedasticity "

Here, we are only interested in adding a third stochastic processes. Several studies (e.g. Langbein 2008, Davis et al., 2012, He et al. 2019) have used the addition of a random-walk component in their stochastic noise model (i.e. White noise + Flicker noise + random-walk). as recalled in line 134-135. The aim is to model small transient signals (e.g. short post-seismic relaxation), residual signals (e.g., due to the non-deterministic nature of the seasonal signal), small offsets (buried in the noise floor) . . . . The sum of all these small amplitude transient signals is included in the definition of our residual time series (line 400-403). The validity of using the random-walk is generally justified in tectonic active areas (Langbein and Svarc, 2019; He et al., 2019). However, we postulate that the use of the Levy processes can generalise the use of the tree stochastic processes model (line 135). We justify our assumptions in line 137-155 (and Table 1).

Our assumptions are based on previous work such as (Williams 2003 and Williams 2004). Therefore, the main assumptions rely on the deviation from the mean for the non-stationarity of the noise, and homoscedasticity in regards to the sum of the two/three stochastic processes. Based on our simulations and experience with GNSS time series, the noise variance is finite (but very large in presence of tails ). Note that we discuss about the finite variance in line 350-360 and appendix D.

(R): We will add the sentence "The main assumptions rely on the deviation from the mean for the non-stationarity of the noise, and homoscedasticity in regards to the sum of the two/three stochastic processes."

- About the use of Levy Flights, the model may be limited for its application to GNSS time series due to the assumptions that GNSS time series have a finite variance. The randomness of the jumps in the GNSS time series is an assumption only valid for small offsets buried in the noise floor, because we cannot detect them by eyes and therefore difficult to model with a step function (See the discussion in appendix D).

- Your comment about looking at non-stationarity beyond the mean using

ARCH/GARCH type of models is interesting for future work (This will be mentioned in the revised conclusions). We have restrained the study on the common assumptions in geodesy with non-stationarity based on the variations of the mean value. It is also worth mentioning that when including a random-walk component, its amplitude is significantly smaller than the other two components (white and coloured noises) – see He et al. (2019).

- Intermittency in GNSS time series has also a limited application. Short high bursts or sudden large deviations for the mean are events which should be modelled (e.g. Earthquake and post-seismic relaxations), unless it is a fault/problem happening at the station. If such event is in the residual time series, it means that there is anxiety between the time series and the functional model. We discuss about this issue when justifying the use of the stable Levy process and the application of the Levy alpha stable distribution (See l. 216-217 and discussion in Section 3.2 – l.359-360).

- "You mention the use of the Hector software at several, in my opinion not necessarily relevant places. Since you do not provide any specific information that is unique to this software, I recommend mentioning this only in the acknowledgements and removing corresponding unnecessary statements elsewhere (e.g. ll. 200-201, 390, 402-403,417-418)"

The Hector software is mentioned and underlined in the manuscript as it was asked in the previous review. The interest of the software is that it is based on the joint estimation of the functional and stochastic models using maximum likelihood. The covariance can be modified to include different types of noise (e.g. White + power-law, White+ Flicker noise, . . .). There are other software used in geodesy using other estimators (e.g. MCMC, Least-squares variance component estimator) which can give different results on the estimated geophysical signals due to different assumptions on the noise model and the stationary properties of the noise in order to limit the computing time (see Montillet and Bos, 2019). That is why as a geodesist, it is important to mention the software and to recall its specificity.

(R): we will limit mentioning the software too much in the revised manuscript.

- Âń Regarding the analysis of the modelling results, I am not convinced that it makes sense statistically to use correlations as a measure for the goodness-of-fit/matching between an empirical and a theoretical distribution. Âż

This approach was chosen in Montillet and Yu (2015). However, we will add the Anderson-Darling test as another measure.

- About the results in Table 3, we need to check them. The large standard deviation associated with the correlation means that there are a lot of variability in the data when applying the distribution. That can be due to the amplitude of the coloured noise with beta equal to 1 or 1.5.

Minor specific comments

Find below the answers to most of your questions from your bullet point list. Note that we will do another grammar check in the revised version.

- "I don't quite see that it is relevant to mention the length of the real-world series at this point"

The length of the time series is important, because it has been shown that it is very difficult to detect random-walk noise in short time series (e.g., L < 9 years). Also, He et al. (2019) showed that in some very long time series (e.g. 9 -10 years), the power spectrum can experience a flattening at high frequencies.

(R): We replace the sentence with : Note that the length of the geodetic time series (L) considered in this study is at least 9years (3285 observations) in order to be able to model correctly the coloured noise and to detect small amplitude random-walk component according to He et al. (2019).

- " Can you explain the relationship between slowly varying mean of colored noise and the Gauss-Markov assumption a bit more explicitly? "

In L. 75 we recalled the Gauss-Markov assumption stating that the noise in GNSS daily position is Gaussian distributed. We assume that the coloured noise variations (around the mean) are slow and not big enough to be able to change the profile of a (multivariate) Gaussian distribution. Thus, that is related to intermittency and aggregation as previously discussed which could skew or completely deform the distribution with such large amplitude events.

(R): Let us add to the sentence "... we assume that the mean of the coloured noise is equal to $\mu C(t)$, slowly varying with time, therefore ruling out the occurrence of specific events of large amplitude such as aggregations or burst of spikes which could invalidate the Gauss-Markov assumption".

- (R): About mentioning the Cauchy-class of processes. We will add in the paragraph (starting L. 90) the following sentence: " Another type of processes worth mentioning is the Cauchy-class of processes, which consist of the stationary Gaussian random processes defined by a correlation function which depends on the Hurst parameter which can be seen as the generalization of some stochastic models (Gneiting and Schlather, 2004). Âż

[Gneiting, T ; Schlather, M (2004) Stochastic Models That Separate Fractal Dimension and the Hurst Effect, 46(2), 269-282, SIAM Review, doi: 10.1137/S0036144501394387]

Note that the relationship between the fBm (Hurst parameter) and the FARIMA is recalled in L 437 in Appendix A.

- About the sentence L.154-155, it comes back to the previous discussion on the Gauss-Markov assumption. Also it is important to underline that in an ideal case where all the geophysical signals are well modelled in the time series, the residual time series should only contain the sum of the different noise components.

In order to create heavy tails distribution, it is most likely due to anxiety in the functional model by forgetting to model large offsets or post-seismic relaxation (e.g. when associated with slow slip events). Perhaps, it can be due also to unknown short-time transient processes. In a special case, it can be also due to the presence of outliers which have not been filtered. Therefore, the GM assumption may not be applied to the distribution of the residual time series.

(R): we will rewrite the sentence such as: "Therefore, the residual time series withholds some remaining unmodelled geophysical signals or unfiltered large outliers which can potentially undermine the Gauss-Markov assumption (e.g., presence of heavy tails in the distribution of the residual time series). Âż

- In L. 262, the driving parameters refer to the parameters of the characteristic function in Eq. 5. We will make it clear by replacing with "the parameters of the characteristic function".

- Ll.271-274: I understand this as that heavy tails in the series can either be attributed to the residuals or to the third component. Can such an attribution be actually unique?

We are not sure what the reviewer means. In order to have heavy tails, you must have large events unmodelled in the residual time series. In GNSS, it is unlikely that the various noise components produce large tails due to the amplitude of the noise.

- L.349: Doesn't the mentioned varying amplitude of the colored noise rather call for multiplicative/heteroscedastic models?

We value the idea of testing heteroscedastic models such as ARCH and GARCH for the future work. As previously said, this study is mainly based on a sum of various noise components. We will mention this future work in the conclusions.

- L.436-439: The link to fBm is a bit unclear here. Since you consider stationary processes (FARIMA class), it would be more reasonable in my opinion to link this to fractional Gaussian noise (fGn) the aggregation of which than provides sample paths for fBm.

(R): we will also mention the representation of the fBm via the fGn in the revised version.

Note that your comments on improving the readability of the manuscript and the better use of the Latex template will be taken into account in the revised manuscript. For example, the appendices will be added as supplementary electronic material.

Thank you for suggesting these improvements.

---

## Author Response (AR1)

[revised manuscript text omitted]

**A.  Geophysical model and ML Estimator**

Following Section 2.1 in order to obtain the residual time series $x(t)$ ($t$ the epoch), we subtract the functional model $s_0(t)$ to the GNSS observations $s(t)$ (see Eq. (1)). The functional model of the geophysical signals is based on the polynomial trigonometric method (Li et al. , 2000; Williams , 2003; Tregoning and Watson , 2009).

$$s_0(t) = at + b + \sum_{j=1}^{N}(c_j \cos(d_j t) + e_j \sin(d_j t)) \tag{A.1}$$

with $s_0(t)$ the sum of the tectonic rate (with coefficient $a$ and $b$) and the seasonal signal (sum of cos and sin functions with coefficients $c_j$ and $e_j$). Note that $d_j$ is equal to $2\pi j/N$, and $N$ can be equal up to 7 (He et al. , 2017). One can also add a Heaviside step function at nominated time $t_i$ in order to estimate the amplitude of an offsets (see appendices). It should be emphasised that the geophysical model is selected based on the surrounding geodynamical activity around the GNSS stations (Montillet and Bos , 2019). Finally, the residual signal is considered to be the remaining geophysical signals (i.e. seasonal component and tectonic rate) not completely estimated due to the mismodelling of the stochastic properties of the time series and other small amplitude (i.e. sub-millimeter) short-time duration transient signals (i.e. local signals, subsidence, ... ) (Bos et al. , 2013; Montillet et al. , 2015; Herring et al. , 2016; He et al. , 2017).

Furthermore in this study, the Hector software is used to estimate jointly the functional and stochastic models in order to produce the residual time series as described in Section 2.4. The software is based on a maximum likelihood estimator (MLE). To recall Montillet and Bos  (2019) (Chapter 2), for linear models, the log-likelihood for a time series of length $L$ can be rewritten as:

$$\ln(Lo) = -\frac{1}{2}\left[L\ln(2\pi) + \ln(\det(\boldsymbol{C})) + (\boldsymbol{s} - \boldsymbol{Az})^T \boldsymbol{C}^{-1}(\boldsymbol{s} - \boldsymbol{Az})\right] \tag{A.2}$$

This function must be maximised. Assuming that the covariance matrix $\boldsymbol{C}$ is known, then it is a constant and does not influence finding the maximum. Next, the term $(\boldsymbol{s} - \boldsymbol{Az})$ represent the observations minus the fitted model and are called the residual time series $\boldsymbol{x}$. Note that $(\boldsymbol{Az})$ is the matrix notation of $s_0(t)$. The last term can be written as $\boldsymbol{x}^T \boldsymbol{C}^{-1} \boldsymbol{x}$ and it is a quadratic function, weighted by the inverse of matrix $\boldsymbol{C}$.

Now let us compute the derivative of $\ln(Lo)$:

$$\frac{d\ln(Lo)}{d\boldsymbol{z}} = \boldsymbol{A}^T \boldsymbol{C}^{-1} \boldsymbol{s} - \boldsymbol{A}^T \boldsymbol{C}^{-1} \boldsymbol{Az} \tag{A.3}$$

The minimum of $\ln(Lo)$ occurs when this derivative is zero. Thus:

$$\boldsymbol{A}^T \boldsymbol{C}^{-1} \boldsymbol{Az} = \boldsymbol{A}^T \boldsymbol{C}^{-1} \boldsymbol{s} \;\rightarrow\; \boldsymbol{z} = \left( \boldsymbol{A}^T \boldsymbol{C}^{-1} \boldsymbol{A} \right)^{-1} \boldsymbol{A}^T \boldsymbol{C}^{-1} \boldsymbol{s} \tag{A.4}$$

This is the weighted least-squares equation to estimate the parameters $\boldsymbol{z}$. Most derivations of this equation focus on the minimisation of the quadratic cost function. However, here we highlight the fact that for observations that contain Gaussian multivariate noise, the weighted least-squares estimator is a maximum likelihood estimator (MLE). Therefore, the Hector software estimates the functional and stochastic parameters via the MLE. Note that in our case $\boldsymbol{C}$ is not a constant, because we assume that the time series contain white and coloured noise. In fact $\boldsymbol{C}$ is equal to the covariance matrix $E\{\mathbf{n}^T \mathbf{n}\}$ in Section 2.1. Thus, the expression of $\boldsymbol{C}$ changes depending on the selection of the stochastic noise model (i.e. Flicker + White noise, Power-law + white noise) discussed in Section 2. Note that further assumptions (i.e. matrix computation) to increase the computational speed can be found in (Bos et al. , 2013) and (Montillet and Bos , 2019). Finally, the Gaussian multivariate noise model (or Gauss-Markov assumption ) holds with the additional assumption that the coloured noise is slowly varying which means that 1/ the non-stationarity of the noise around the mean, and 2/ no intermittency in the time series, i.e. no events creating short high bursts or sudden large deviations for the mean.

**B.   Relationship between the FARIMA, ARMA and fBm**

Following Granger and Joyeux  (1980), Panas  (2001) and Pipiras and Taqqu  (2017), a time series (e.g., $\mathbf{x}$) follows an FARIMA $(p,d,q)$ process if it can be defined by

$$
\begin{aligned}
\Psi_p(Z)x(t) &= \Theta_k(Z)(1-Z)^{-d}b(t) \\
\Psi_p(Z) &= 1 - a_1 * Z - a_2 * Z^2 - ... - a_p * Z^p \\
\Theta_q(Z) &= 1 + b_1 * Z + b_2 * Z^2 + ... + b_k * Z^q \\
(1-Z)^{-d} &= \sum_{j=0}^{\infty} \frac{\Gamma(j+d)}{\Gamma(j+1)\Gamma(d)} * Z^j
\end{aligned}
\tag{B.5}
$$

where $E\{\mathbf{b}\}$ equal zero and $\sigma_b^2 < \infty$. $Z$ is the backshift operator applied to $x$ $(t)$ and $b$ $(t)$, respectively. The properties of the FARIMA model are presented by Granger and Joyeux  (1980): i) if the roots of $\Phi_p(Z)$ and $\Theta_q(Z)$ are outside the unit circle and $d < |0.5|$, then $\mathbf{x}$ is both stationary and invertible; ii) if $0 < d < 0.5$ the FARIMA model is capable of generating stationary

series which are persistent. In this case the process displays long-memory characteristics, with an algebraic autocorrelation decay to zero; iii) if $d \geq 0.5$ the process is non-stationary ; iv) when $d$ equal to $0$ it is an ARMA process exhibiting short memory; v) when $-0.5 \leq d < 0$ the FARIMA process is said to exhibit intermediate memory or anti-persistence. This is very similar to the description of the Hurst parameter in the fBm model. Note that one can underline the relationship between $d$ and $H$ such as $H = d + 0.5$, well-known in financial time series analysis in the presence of aggregation processes (Panas , 2001). Note that, the fBm can be defined by its integral representation (see supplementary material - $C$) or the aggregation of the fractional Gaussian noise (Taqqu et al. , 1995).

**C.   fBm and fLsm: integral representation and discussion**

The fractional Brownian motion (fBm) $\{B_H(t)\}_{t \geq 0}$ has the integral representation:

$$B_H(t) = \int\limits_{-\infty}^{\infty} \left((t-u)_+^{H-\frac{1}{2}} - (-u)_+^{H-\frac{1}{2}}\right) dB(u) \tag{C.6}$$

where $x_+ = max(x, 0)$ and $B(u)$ is a Brownian motion (Bm). It is $H$-self-similar with stationary increments and it is the only Gaussian process with such properties for $0 < H < 1$ (Samorodnitsky and Taqqu , 1994). It is worth mentioning that a damped version of the fBm exists and known as the Matérn process, defined having a sloped spectrum that matches fBm at high frequencies and taking on a constant value in the vicinity of zero frequency (Lilly et al. , 2017). However, this process is out of the scope of this study.

From Weron et al. (2005), the fractional Lévy stable motion (fLsm) can be defined with the process $\{Z_\alpha^H(t)\}$ (with $t$ in $\mathbb{R}$) by the following integral representation:

$$\begin{aligned} Z_\alpha^H(t) &= \int\limits_{-\infty}^{\infty} \left((t-u)_+^{H-\frac{1}{\alpha}} - (-u)_+^{H-\frac{1}{\alpha}}\right) dZ_\alpha(u) \\ &= \int\limits_{-\infty}^{\infty} f_t(u) dZ_\alpha(u) \end{aligned} \tag{C.7}$$

where $Z_\alpha(u)$ is a symmetric Lévy-stable motion (Lsm). The integral is well defined for $0 < H < 1$ and $0 < \alpha \leq 2$ as a weighted average of the Lévy stable motion $Z_\alpha(u)$ over the infinite past with the weight given by the above integral kernel denoted by $f_t(u)$. The process $Z_\alpha^H(t)$ is $H$-self-similar and has stationary increments. Note that $H$-self-similarity follows from the above integral representation and the fact that the kernel $f_t(u)$ is $r$-self-similar with $r = H - 1/\alpha$, when the integrator $Z_\alpha(u)$ is $1/\alpha$-self-similar. This implies the following important relation:

$$H = r + 1/\alpha \tag{C.8}$$

The representation Eq. (C.7) of fLsm is similar to the representation (C.6) of the fractional Brownian motion. Therefore fLsm reduces to the fractional Brownian motion if one sets $\alpha = 2$. When we put $H = 1/\alpha$ we obtain the Lévy $\alpha$-stable motion

which is an extension of the Brownian motion to the $\alpha$-stable case. At the contrary to the Gaussian case ($\alpha = 2$) the Lévy $\alpha$-stable motion ($0 < \alpha < 2$) is not the only $1/\alpha$-self-similar Lévy $\alpha$-stable process with stationary increments. This is true for $0 < \alpha < 1$ only (Weron et al. , 2005). Note that this definition of the Fractional Lévy process is different from Benassi et al. (2004), which is not a self-similar process. Finally, another type of processes worth mentioning is the Cauchy-class of processes, which consist of the stationary Gaussian random processes defined by a correlation function which depends on the Hurst parameter which can be seen as the generalization of some stochastic models (Gneiting and Schlather , 2004).

**D. Some Considerations on the Simulations of the Mixed Noise in the GNSS time Series**

This section recalls one way to simulate the coloured noise in the GNSS time series following Montillet and Bos (2019). Granger and Joyeux (1980) and Hosking (1981) demonstrated that power-law noise can be achieved using fractional differencing of Gaussian noise:

$$(1 - B)^{-K/2} \boldsymbol{v} = \boldsymbol{w} \tag{D.9}$$

where $B$ is the backward-shift operator ($Bv_i = v_{i-1}$) and $\boldsymbol{v}$ a vector with independent and identically distributed (IID) Gaussian noise. Hosking and Granger used the parameter $d$ for the fraction $-K/2$ which is more concise when one focuses on the fractional differencing aspect. However, in Geodesy the spectral index $K$ is used in the equations. Hosking's definition of the fractional differencing is:

$$
\begin{aligned}
(1 - B)^{-K/2} &= \sum_{i=0}^{\infty} \binom{-K/2}{i} (-B)^i \\
&= 1 - \frac{K}{2} B - \frac{1}{2} \frac{K}{2} \left(1 - \frac{K}{2}\right) B^2 + \dots \\
&= \sum_{i=0}^{\infty} h_i
\end{aligned}
\tag{D.10}
$$

The coefficients $h_i$ can be viewed as a filter that is applied to the independent white noise. These coefficients can be conveniently computed using the following recurrence relation (Kasdin , 1995):

$$
\begin{aligned}
h_0 &= 1 \\
h_i &= \left(i - \frac{K}{2} - 1\right) \frac{h_{i-1}}{i} \quad \text{for } i > 0
\end{aligned}
\tag{D.11}
$$

One can see that for increasing $i$, the fraction $(i - K/2 - 1)/i$ is slightly less than 1. Thus, the coefficients $h_i$ only decrease very slowly to zero. This implies that the current noise value $w_i$ depends on many previous values of $\boldsymbol{v}$. In other words, the noise has a long-memory. Eq. (D.11) also shows that when the spectral index $K = 0$, then all coefficients $h_i$ are zero except for $h_0$. This implies that there is no temporal correlation between the noise values. One normally assumes that $v_i = 0$ for $i < 0$.

With this assumption, the unit covariance between $w_k$ and $w_l$ with $l > k$ is:

$$C(w_k, w_l) = \sum_{i=0}^{k} h_i h_{i+(l-k)} \tag{D.12}$$

100    Since $K = 0$ produces an identity matrix, the associated white noise covariance matrix is represented by the unit matrix $\boldsymbol{I}$. The general power-law covariance matrix is represented by matrix $\boldsymbol{J}$. The sum of white and power-law noise can be written as (Williams , 2003).It is a widely used combination of noise models to describe the noise in GNSS time series (Williams et al. , 2004). Besides the parameters of the linear model (i.e. the functional model), maximum likelihood estimation can be used to also estimate the parameters $K$, $\sigma_{pl}$ and $\sigma_{wn}$. This approach has been implemented various software packages including
105    Hector (Bos et al. , 2013).

We assumed that $v_i = 0$ for $i < 0$ which corresponds to no noise before the first observation. This is an important assumption that has been introduced for a practical reason. For a spectral index $K$ smaller than $-1$, the noise becomes non-stationary. Most GNSS time series contain flicker noise which is just non-stationary. Using the assumption of zero noise before the first observation, the covariance matrix slowly grows over time but always remains finite.

**Rebuttal letter for the manuscript: Application of Lévy Processes in Modelling (Geodetic) Time Series with Mixed Spectra**

Dear Editor, reviewers,

Thank you for your comments. We have modified the manuscript based on the comments of the reviewers. During the discussion phase, we did reply to the major comments and we have included some suggestions that we have followed for this revised manuscript.

The major modifications are:

- Section 2.1 has been rewritten based on R1 comments
- The N-steps algorithm is described with more details.
- The simulations of the GNSS time series have been revised to include the Anderson-Darling test as suggested by R2
- We have also improved the Figures via simulations and also reduced the steps in the variation of the length of the time series (which also improved the Figures with real time series).
- We have also applied the Anderson-Darling test to the real time series
- As R2 pointed out, we have put some of the appendices in the "supplementary material". Thus, Appendix A, B, C and E are treated as supplementary material (as A, B, C, D). Only D and F should remain as appendices (as Appendix A, Appendix B). The supplementary material is dissociated from the manuscript to comply with the NPG guidelines.

Finally, R2 has pointed out an interesting research direction with the ARCH/GARCH model which will be the topic of our next work.

Thank you for your comments.

[RC1/SC1]

**Major comments**

- *« The mathematical model is really messily written, and it is hard to understand it. Take all the definitions from the appendix, and put them in the text, and remove all thespeculative/descriptive material. »*

  (R). As underlined in the discussion (SC1), there are a few points which must be emphasized when working with geodetic data. Thus, we have rewritten the whole Section 2.1 to include the comments from R1. Also, we reduced the descriptive material in this section.
  Some more details are now in the Supplementary material A.

- *"After you have done the basic model, you can construct an a posteriori distribution of the unknown parameters, and then the target is to sample these parameters withMCMC methods or obtain optimisation-based MAP/ML estimators. Define explicitly your estimators with respect to posterior distribution. Of course you can use some other constructions as well, but define your estimators explicitly. The N-step method is totally heuristic and should not be included in the manuscript. Please come up with some mathematics/stats-based parameter estimation algorithm."*

(R) We have discussed our vision in the SC1 comprehensively. However, the idea of the reviewer is similar to what R2 pointed out for future work, on investigating the use of the ARCH model. We believe that is valuable and interesting approach for a future research. We have taken this in the revised manuscript (see the conclusions).

Note that the N-step method is further explained. It emphasises that we use the ML estimator (as highlighted in section 2.1) to estimate jointly the stochastic and functional model.

**Minor Comments**

l.27 long-memory processes > deleted – see sentence

l. 43 the sentence now reads: "The Levy stable distributions can model the heavy tail characteristics of some data sets with generally infinite variance."

p.3 Eq. 3
Section 2.1 including Eq. (1) is rewritten completely. See modifications.
Also, we consider the noise "n" as a continuous process, with mean equal to the coloured noise (as the white noise is zero mean by definition).

p.3
 The covariance of the coloured noise "J" is named following the literature (see e.g., Williams et al. 2004 [eq. 4], Bos et al. Chapter 2 [eq. 27] in Montillet and Bos 2019).

p.3 ""Note that the length of the geodetic time series (L) considered in this study is at least 9 years ( 3285 observations)." Can you plot this data"

Several examples of time series are given with the real time series in Section 3.1.2 and Appendix B (additional figures)

p.3 " …which states that the noise is Gaussian distributed, therefore n follows .." > corrected – now it reads " …which states that the noise is Gaussian distributed.

p.3 L82 "p. 3: Line 82 – You have not defined β, and for all β <0, the matrix J(β)does not exist. Be more precise."

This section has been revised, where β is now K in [0, 2]. Note that the definition of K with the different values is given after Eq. 4.

Thank you for your nice comments.

[RC2/AC1]

**Major comments**

1/ *"Can you explain a bit more systematically how (and why) different types of non-stationarity are associated to the different components of your stochastic model?"*

(R): According to the discussion (RC2), we have added the sentence (l.73-74): "The coloured noise results from various parameters during the processing of the GNSS observations such as the mismodelling of GNSS satellites orbits, … "

2/ *About the length of the time series*

(R) we modified the text according to the RC2 (l 93-94): "Note that the value of L is here at least 9 years (3285 observations) in order to model correctly the long-range dependencies associated with the coloured noise and to detect slow transient signals ..."

3/ *"I am wondering about non-stationarity beyond just the mean, i.e., possible time-dependence inthe variance (or even higher-order distributional characteristics), which I believe can-not be captured by the present model setup but would require either some multiplicative component (e.g., functional model times noise of a certain type)or a stochastic model component beyond ARMA/FARIMA that inherets the property of conditional heteroscedasticity "*

(R) We discuss it in the RC2. We added some more information in the sentences (Supplementary material A – last sentence) "Finally, the Gaussian multivariate noise model (or Gauss-Markov assumption - see Section 2.1) holds with the additional assumption that the coloured noise is slowly varying which means that 1/ the non-stationarity of the noise around the mean, and 2/ no intermittency in the time series, i.e. no events creating short high bursts or sudden large deviations for the mean."

- *About the use of Levy Flights*

As discussed in the RC2, it is difficult to include this model within GNSS time series, due to the assumption that GNSS time series have a finite variance and the assumptions that the small amplitude transient signals should not generate jumps larger than the noise floor. Therefore, we have left this model out for now.

- *Intermittency*

Following the discussion in RC2, it is mentioned in the revised manuscript in the assumptions related to the Gauss-Markov model (see added sentence above and in Section 2.1)

- *Hector Software:*
It is only mentioned once in the main text. The use of the software to decrease the log-likelihood function (via MLE) is described in the supplementary material (A). We also added an acknowledgment.

- *Homoscedasticity vs Heteroscedasticity and the ARCH/GRACH model*

(R) We have discussed in the RC2 that our model is based on homoscedasticity with the sum of two (three) stochastic processes. However, we would like to circumvent a discussion on this property. We believe that it will be more appropriate on a future work about the GARCH model. We have added at the end of the conclusions: "Finally, future work should investigate the autoregressive conditional heteroscedasticity (ARCH) model applied to GNSS time series in order to model differently the stochastic properties (e.g. non-stationarity beyond the mean)."

- *The Anderson-Darling Test:*

(R) About the results in Table 3 and 4 (and A.1) using the correlation coefficient. As mentioned in the header of this letter, we have implemented the Anderson-Darling test. Therefore, the results section is revised accordingly. It supports the previous results where the Levy alpha stable distribution mostly detect high tails for the last scenario (c).

**Minor comments:**

- We change "soil motion" by "ground motion"

- The justification of the length of the time series has been added as discussed above.

- *" Can you explain the relationship between slowly varying mean of colored noise and the Gauss-Markov assumption a bit more explicitly? "*

  (R) According to the discussion, we revised the paragraph L95-100.

- *About the Cauchy-class of processes and H*

  In addition of the discussion in RC2, we have added in the supplementary material (C) : "Finally, another type of processes worth mentioning is the Cauchy-class of processes, which consist of the stationary Gaussian random processes defined by a correlation function which depends on the Hurst parameter which can be seen as the generalization of some stochastic models … "

- *In the context of this brief discussion of FARIMA models …*

  (R) The relationship between FARIMA and H is discussed in the supplementary material (B)

- *The definition includes a variable k that I don't find appearing anywhere before.*

  (R) In the revised manuscript, "k" is now "beta", which is the symmetry parameter in eq. (5)

- *About L154-155:*
  As discussed in the RC2, we have modified the sentence such as:
  "Therefore, the residual time series withholds some remaining unmodelled geophysical signals or unfiltered large outliers which can potentially undermine the Gauss-Markov assumption (e.g., presence of heavy tails in the distribution of the residual time series) "

- *L.171: In which sense do you consider colored noise to be non-stationary*

  (R) as discussed in RC2, the coloured noise is here non-stationary around the mean. This sentence is revised.

- *L.243: The list of values given in the text is inconsistent with that shown in the figure.*

  (R) there is a confusion with the figures in the previous version. Now "K" is the power-law exponent and it is the same in the figures and the text.

- *L.246: I would not speak of "earlier" here due to the low number of data points in the figure, but rather refer to the overall values of the variance.*

  (R) The sentence is revised accordingly.

- *L. 262 "the driving parameters "*

  (R) replaced with "the parameters of the characteristic function"

- *L.271-274: I understand this as that heavy tails in the series can either be attributed to the residuals or to the third component. Can such an attribution be actually unique?*

(R) As discussed in RC2, it is not clear what the reviewer means. It seems the question is clearly given. Why not understand?

- *L.325 and several times later: There are quite a few cases of equations spanning …*

(R) We have corrected this issue, eliminating the right hand-side.

- *L.348: "colored noise can generate long-memory processes" seems a bit odd to me*

(R) we revised the sentence: "coloured noise is characterised by long-memory processes"

- *L.349: Doesn't the mentioned varying amplitude of the colored noise rather call formultiplicative/heteroscedastic models?*

(R) We value the idea of testing heteroscedastic models such as ARCH and GARCH for the future work. As previously said, this study is mainly based on a sum of various noise components. We have added a sentence in the conclusions.

- L.373: *"long-term correlations"* > corrected

- L.434: instead of the term "hyperbolic", "algebraic" seems more commonly used " > corrected

- *Ll.417-430: Please emphasize that the expressions in Z denote composite operators in terms of the backshift operator Z applied to x(t) and b(t), respectively* > corrected

- *L.436-439: The link to fBm is a bit unclear here. Since you consider stationary pro-cesses (FARIMA class), it would be more reasonable in my opinion to link this to fractional Gaussian noise (fGn) the aggregation of which than provides sample paths for fBm.*

(R) We added "Note that, the fBm can be defined by its integral representation (see supplementary material - C) or the aggregation of the fractional Gaussian noise … "

- *L.451: extend the equation by an expression including f_t(u) as the latter is used in the text below*

(R) We added the kernel f_t(u) in Eq. (C.7) in the supplementary material.

Finally, we have followed your editorial recommendations by checking the languages and moving some of the appendices into the supplementary material.

---

## Author Response (AR2)

**Letter to the Reviewer**

Dear Editor, Dear Reviewer,

Thank you for the comments. The manuscript has been modified following your suggestions. Please, find some details to your major comments. Note that the minor points are corrected in blue in the text, in addition of the added sentences.

Thank you for the additional improvements.

*Major comments raised by the reviewer*

1/ ... "What I however should request is adding some cautionary remarks regarding the feasibility of using correlations for the purpose for which they are used here."

**R: We agree with the reviewer as already mentioned in the previous round where R2 suggested to add the Anderson-Darling Test for the same reason. Therefore, we add (p. 11 l. 286-288):**
*"However, we acknowledge that the (Pearson) correlation coefficient could be biased due to inherent normalized sum constraint between the distributions estimated directly from the data. Therefore, in this instance the AD test should be more reliable."*

2/ ".However, in discussing the outcomes of this test, the authors frequently speak about an "accepted null hypothesis", which is a heavily flawed phrasing."

**R: This term is loosely used in documentation of statistical tests described in various software (e.g. MATLAB). We agree with the reviewer and changed "accepted null hypothesis" by "not rejected null-hypothesis" as suggested in captions of Tab. 3, 4 and A1, text in ll. 279, 287, 329**

Minor comments:

1. Separating main text, appendices and supplementary information has greatly improved the readability of the manuscript. Given that a concise paper is most appealing for readers, I would even suggest moving the additional Tab. A1 and Figs. A1-A4 to the supplementary material, since they do not add essential information to the manuscript and in fact just blow up its size (thereby also increasing the page charges to be calculated by the publisher). My recommendation would be keeping only Appendix A in the main document, possibly not even using a numbering then and referring to its content in the main body simply to as "see Appendix" or similar.

**R: We have kept only appendix A in the main document. Appendix B is now Section E in the supplementary material. The reference in the main text mentions only "Appendix".**

2. In order to improve the internal referencing of the material, I strongly recommend the authors to revise references to the supplementary material by emphasizing explicitly the part of that document that is referred to, e.g., "see Supplementary Material Sect. A" or "Supplementary Material Fig. S1/Tab. S1" if following my suggestions under the previous point 1.

**R : Done**

3. Please use consistently the term "Gaussian Lévy" (or "Lévy Gaussian", but I would tend to prefer the former) instead of mixing both terms.

**R : We homogenized the manuscript following the remark (Gaussian Lévy).**

4. The block of equations on pp.7-8 misses a few equation numbers, including such that are later referred to in the manuscript (e.g., Eq. (7)). This should be clarified.

**R : We have added one number for each block to avoid unnecessary notation. The manuscript is revised accordingly**.

5. I experienced some personal confusion with the authors' use of the term "epoch". Would you mind explaining its meaning in the context of this work precisely along with its first occurrence (p.3, below Eq. (1))?

**R : « epoch » means the time associated with an observation. We have clarified the meaning by replacing "L epochs" with "L observations", and also using "time step" in the definition of the multivariate noise model below eq. (2).**

6. Ll.19-20: As it is written now, one may think that Flicker noise and white noise are examples of band-pass noise, which I would clearly object. Please rephrase.

**R : The sentence is rephrased – see in the introduction "To name a few …"**

7. L.177: "nonstationary signal (around the mean)" – do you mean a signal with a non-stationary mean or (as I suppose) a signal with a stationary (zero?) mean and nonstationary variations about this mean?

**R : The non-stationary properties of the coloured noise involve that 1/ the mean value varies with time, and 2/ there are variations around the mean. In other words, depending on the power-law exponent, the phenomena of aggregation or intermittency can exist. However, due to the small amplitude of the coloured noise in the GNSS time series, we "do" exclude large variations around the mean, therefore we use "non-stationary (around the mean)". This is explained in the supplementary material Section A – at the end. We have added the reference in the main text.**

8. L.215: "see also the discussion on the Hector Software (Bos et al., 2013) in the Supplementary Material Sect. A).

**R : Done**

9. L.231: p and q are NOT lags, but the AR and MA model orders of the considered model.

**R : it is replaced: lags => model orders**

10. L.372: "uncorrelated assumption" => "assumption of uncorrelated components"

**R : Done**

11. Eq. (A.2), first line: the meaning of the brackets is not clear If placed as now, they are not necessary.

**R : We are not sure what the reviewer means**. **This is the same definition as Montillet and Bos, 2019 – Chapter 2.**

 In Eq. (A.3), the brackets however appear necessary, but you should use \left( and \right) to enforce brackets of a proper size for the enclosed content.

**R : It is revised – see Eq. (A.3) and (A.4)**

12. Supplementary Material Section A: The text in ll.16-22 is a literal repetition of ll.84-90 of the main manuscript. Just shorten this to one sentence with reference to the corresponding Eq. (3).

**R : the paragraph is shortened accordingly**

**R: Please, note that all the following corrections are written in blue in the manuscript when it is appropriate**

• L.8: "The fractional Lévy process…"
• L.9: "The stable process…"
• L.10: "Therefore, it implies…" – what does "it" refer to**? R : the model**
• L.15: "phenomena"
• L.15: "earthquakes"
• Ll.16-17: "These time series provide the estimated daily positions of…"
• L.19: "describe"
• L.20: "cyclic"
• L.20: "referred to"
• Ll.27-28: "To name a few, previous choices include the…"
• L.39: "distributions"
• L.40: "from a Gaussian distribution"
• L.47: "to the residual…"

• L.77: "a multivariate continuous-time stochastic process"
• L.80: "transposition operator"
• L.81: "$K\in [0,2]$"   **R: "0" is excluded as it is the case of white noise, which is defined by the identity matrix in the first term.**

• L.88: "represents"
• L.89: "model for x" **R: better "also called X in Eq …"**
• L.93: "correctly" might better read "properly" (there is not ground truth…)
• L.98: "bursts"
• L.107: "a random walk"
• L.109: use \citet instead of \citep for reference Eke et al.(2002)
• L.114: "based on different"
• L.123: close bracket not after "2018", but after "1.6" **R: we added [ .. ]**
• L.127: "where sign(.)…"
• L.128: "sets"
• L.132: use \citet instead of \citep for reference Weron et al. (2005)
• L.133: "with $H$ being"
• L.140: "similar approach as used"
• L.144: "Gaussian Lévy"
• L.146: "This is the special case"
• L.152: "ratio between the amplitudes of the coloured and white noise components determines…"
• L.153: "most suitable among the FARIMA…"
• L.157: "modelled"
• L.167: "as a random walk"
• L.220: "Gaussian Lévy"
• L.233: "selection (ARMA, FARIMA)"
• L.234: "(e.g., Panahi (2016))"
• L.238: "than that of the white noise… should be oreferred de facto...."
• L.245: "power-law noise"
• L.250: "between 1 and 3" **R: we stick with the maths' notations.**

• Ll.254-256: don't put physical units in math mode (no italics). **R: That was asked by previous reviewer**
• Ll.257, 269: "in scenario C"
• L.272: "the fits"
• L.273: "over 50 simulations"
• L.274: "the average correlation"
• L.275: I suggest not to capitalize "normal" (also in what follows from here) -**R: we want to be consistent with the whole text**
• L.282: do not capitalized "intermediate" -**R: we want to be consistent with the tables**

• L.282: "However, scenario C…."
• L.283: "introduce"
• L.284: "performs"
• L.286: "displayed"
• L.287: "…in Table 2 give the probability…"
• L.297: "of the three GNSS stations DRAO,…" (note that I would also suggest not to put the station abbreviations in italics, but this is up to the authors…)
• L.309: "dependence on the…"

• L.320: "explained by…"
• L.321: use \citep instead of \citet for reference to He et al. (2019)
• L.323: "series of ASCO**" R: We do not see the issue**
• L.326: "of the noise that is much larger…"
• L.327: "is approximately the same"
• Ll.332,333: "Up coordinate"
• L.341: "develop closed-form expressions for the mean…."
• L.344: "equal to 1"
• Fig. 3, caption: "function of the residual" (wo times, also in the captions of the other similar figures currently placed in the appendix)
• L.351: "intersect" => "intercept"?
• L.356: remove "Note that" (not necessary) and "if" (grammatically useless)
• L.358: the inline equation seems to miss brackets around the content that is summed up
• L.368: "where Cross…"
• L.371: "with the residual signals"
• L.392: remove "exhibiting"
• L.393: "to using"
• L.398: "an N-step"
• L.417: "in the presence of remaining offsets can be written as"
• L.431: remove "note that"
• SI, l.3: "from the GNSS…"
• SI, l.8: what do you mean by "nominated time"? **R : changed with " at time t.."**
• SI, l.14: remove "Furthermore"
• SI, l.33: remove "Note that"
• SI, l.53: "aggregation of fractional…"
• SI, l.69: remove "Note that"
• SI, l.73: "The representation of fLsm in Eq. (C.7) is similar…"
• SI, ll.101-102: The sum… can be written as…" – this sentence is incomplete. **R: the sentence is deleted**